# GraphCPD: Coherent Point Drift for Point Cloud Registration via Graph Signal Processing

## Abstract

Probabilistic point cloud registration has attracted increasing attention due to its robustness to noise, outliers and occlusions. However, existing methods often suffer from high computational cost and neglect the role of informative priors. In this paper, we propose a new probabilistic registration method based on graph signal processing (GSP), called graph coherent point drift (GraphCPD). Specifically, we design a high-pass graph filter to extract high-frequency components, which are theoretically proven to be invariant under rigid transformations. These components are combined with point coordinates and normals to form a high-dimensional graph signal. We construct a local graph based on the graph signal and use the graph Laplacian model for registration. Compared with the classical Gaussian mixture models (GMMs), graph Laplacian models provide more discriminative geometric representations and enhances the model's ability to capture graph structure. Furthermore, we exploit the invariance of high-frequency components to define prior probabilities, significantly reducing the corresponding search space and improving the speed of registration. Experimental results demonstrate that our method improves runtime efficiency over most existing probabilistic methods, while maintaining competitive registration accuracy, especially on large-scale point clouds. The source code is available at https://anonymous.4open.science/r/GraphCPD-801E.

## 1 Introduction

Point cloud registration is a fundamental task in computer vision and 3D data processing, aiming to align two sets of point clouds and estimate the spatial transformation between them. It plays a crucial role in various applications such as SLAM (Yang et al., 2020), object pose estimation (Jiang et al., 2023), dense 3D reconstruction (Whelan et al., 2016) and interactive tracking of articulated (Mundermann et al., 2007). However, registration remains a difficult problem due to challenges such as noise, outliers, occlusions, partial overlap and varying point densities. The Iterative Closest Point (ICP) algorithm is the most widely used approach, achieving alignment by iteratively finding nearest-neighbor correspondences between points (Chetverikov et al., 2002; Serafin & Grisetti, 2015). Despite its popularity, the ICP algorithm is sensitive to noise and outliers, and may be trapped in local optima (Myronenko & Song, 2010).

Probabilistic methods model point cloud registration as a task of estimating probability density (Jian & Vemuri, 2010; Stoyanov et al., 2012; Gao & Tedrake, 2019). They typically assume that the observed data points follow a normal distribution around the model points, thereby transforming the point-to-point correspondence problem into the estimation of mixture model parameters. These approaches exhibit strong robustness by incorporating an outlier component and spherical isotropic covariance into the GMM (Myronenko & Song, 2010; Evangelidis et al., 2014; Horaud et al., 2010). However, such methods are often computationally expensive and significantly slower than ICP-based approaches, particularly when dealing with large-scale or high-resolution point clouds.

In this paper, we propose a novel probabilistic registration method integrated with GSP providing practical robustness and computational efficiency. We employ high-pass graph filtering to extract high-frequency components, effectively suppressing low-frequency signals and emphasising local

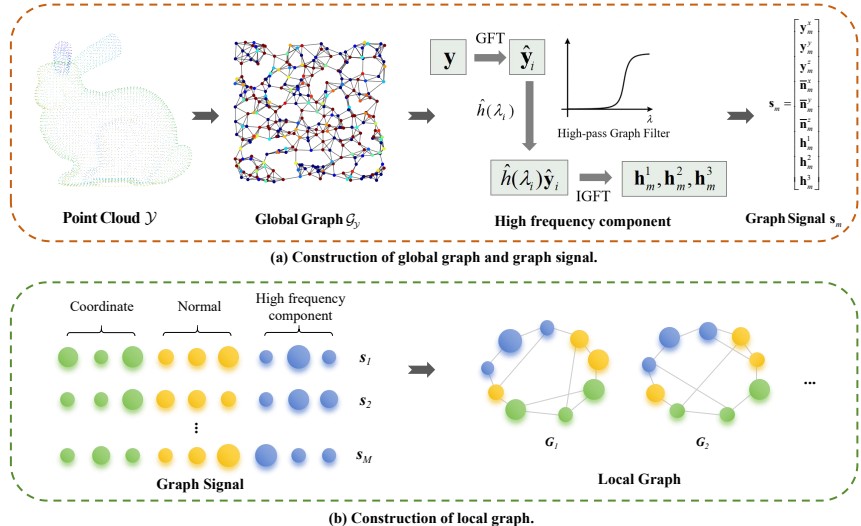

Figure 1: An illustration of the GraphCPD method. (a) Global graph construction with multi-dimensional components as graph signals. GFT denotes graph Fourier transform and IGFT denotes inverse graph Fourier transform. (b) Local graph construction, where graph signals are smoothed on the respective graph Laplacian matrix.

geometric variations. These high-frequency components are theoretically proven to be invariant under rigid transformations, making them particularly suitable for point cloud registration. We construct a high-dimensional graph signal for each point by concatenating the extracted components with its geometric attributes (coordinates and normals). Then, a local graph is constructed for each point based on the graph signal, with edge weights encoding the similarity between different attributes. Based on this, the isotropic covariance matrices commonly used in GMM are replaced with point-specific Laplacian matrices, providing a more geometrically meaningful representation. Furthermore, the cost function is extended from relying solely on coordinates to high-dimensional graph signals, thereby leveraging richer information during the registration process. The maximum likelihood estimation is optimized through the Expectation-Maximization (EM) algorithm (Bishop & Nasrabadi, 2006). Finally, we leverage the extracted high-frequency components to design prior probability, which reduces the corresponding search space and improves the efficiency of the registration process.

The contributions of this paper are are three-fold. **(i)** We introduce a representation dimensionality expansion from points to graph signals for each point in the point cloud. High-frequency components are extracted using a high-pass graph filter, and their invariance under rigid transformations is established (Theorem 1). These components are then concatenated with the coordinates and normals to construct high-dimensional graph signals. Integrating these graph signals into the registration process introduces more geometric detail and enhances registration accuracy. **(ii)** We propose a novel graph-based component-wise density function. For each point, we construct a local graph Laplacian matrix based on its graph signals, whose pseudo-inverse replaces the traditional isotropic covariance matrix in the GMMs. This substitution not only preserves the classical objective of minimizing point-wise discrepancies but also requires their variation trends in the joint representation of multiple attributes within the local graph structure to remain consistent, thereby effectively incorporating graph-structural characteristics. **(iii)** An effective prior probability construction method is proposed. Unlike traditional probabilistic methods that ignore prior information, our approach leverages the extracted high-frequency components to construct prior probability. This allows each point to consider only a limited subset of candidate correspondences, which significantly accelerates the registration process, especially when dealing with large-scale point clouds. To the best of our knowledge, no existing probabilistic based method has yet considered the issue of prior probability.

## 2 RELATED WORK

This paper mainly reviews the probabilistic registration methods based on GMM and the registration methods using GSP tools. Probabilistic methods assume that the distribution of the point set can be modeled as a mixture of Gaussians, and they estimate soft correspondences between two point sets in a probabilistic manner (Ravikumar et al., 2017; Eckart et al., 2018; Gao & Tedrake, 2019; Granger & Pennec, 2002; Horaud et al., 2010; Luo & Hancock, 2003; McNeill & Vijayakumar, 2006; Min et al., 2019b; Myronenko & Song, 2010). Among classical probabilistic registration frameworks, Coherent Point Drift (CPD) is one of the most representative approaches (Myronenko & Song, 2010). It treats the target point set as centroids of a GMM and maximizes the likelihood of the source points via the EM algorithm. While CPD is highly robust, its major drawback is its high computational cost, as it requires the calculation of pairwise probabilities for all points in each iteration. To address this limitation, FilterReg employs Gaussian filtering to approximate the soft correspondence computation in the E step (Gao & Tedrake, 2019). This leads to a substantial speedup while preserving high levels of accuracy and robustness. However, FilterReg relies on complex filter construction and approximation, which increases implementation and integration costs. Additionally, using Gaussian filters for probability estimation can introduce numerical approximation errors. The ECMPR update allows for general anisotropic covariance by updating the entire covariance matrix in the M step (Horaud et al., 2010; Min et al., 2019a). While it provides greater flexibility in covariance modelling and more straightforward handling of outliers, its increased algorithmic complexity and slower convergence rate restrict its use to large-scale scenarios. LSG-CPD improves registration accuracy by adaptively adding different levels of point-to-point penalties based on local surface flatness and computes on GPU to improve speed (Liu et al., 2021). However, it still relies on exhaustive pairwise traversal, resulting in limited computational efficiency for large-scale point clouds.

With the continuous development of GSP theory, its advantages in handling non-Euclidean structured data have become clearer, making it naturally applicable to the analysis and processing of irregular data such as point clouds. Bastico et al. (2024) propose a novel technique based on Laplacian eigenmap coupling, which performs coarse registration by explicitly modeling local structures. GraphReg leverages GSP to extract local geometric features and incorporates them into a rigid-body dynamics framework for mass and force modeling, thereby enabling high-precision and computationally efficient registration (Zhao et al., 2022). However, when establishing correspondences based on the invariance of graph structures, GraphReg adopts a one-to-one hard matching strategy, which is susceptible to outliers or local perturbations, potentially affecting the stability and robustness of the registration process.

## 3 METHOD

### 3.1 FORMULATION OF PROBABILISTIC MODEL

We denote the target set by $\mathcal{Y} = \{\mathbf{y}_m\}_{1 \le m \le M}$, and the source set by $\mathcal{X} = \{\mathbf{x}_n\}_{1 \le n \le N}$. Each point in the source set undergoes a rigid transformation. The 3D coordinates of the transformed points are parameterised by $\mathbf{\Theta} = \{\mathbf{R}, \mathbf{t}\}$, where $\mathbf{R}$ and $\mathbf{t}$ denote the rotation matrix and translation vector, respectively. To estimate the registration parameters, it is necessary to establish the correspondences between the target points and the source points. In classical GMM-based point cloud registration methods, each target data point $\mathbf{y}_m$ is assigned to a Gaussian component centered at the transformed source point. The covariance matrix $\mathbf{\Sigma}_m$ in GMM-based method is typically chosen to be isotropic.

Our method lifts each point in the point cloud from 3D spatial coordinates to a high-dimensional graph signal. This graph signal consists of spatial coordinates, surface normals and high-frequency components extracted via high-pass graph filtering (Fig. 1(a)). Then, we construct a local graph for each point, where the edge weights encode the similarity between the attributes of each dimension of the graph signal. These graph signals exhibit smoothness over their respective local graphs, as illustrated in Fig. 1(b). When the target point $\mathbf{y}_m$ is correctly aligned with the source point $\mathbf{x}_n$, the transformed source signal should preserve smoothness on the graph $\mathbf{G}_m$ constructed from $\mathbf{y}_m$, and vice versa. Motivated by this mutual smoothness property, we define the following component-wise density function:

$$p(\mathbf{s}_n \mid m) = c_m \exp\left(-\frac{1}{2\sigma^2} \|\mathbf{s}_m - \mathcal{T}(\mathbf{s}_n)\|_{\tilde{\mathbf{L}}_m}^2\right). \tag{1}$$

Here, $\mathcal{T}(\mathbf{s}_n) = \mathrm{diag}(\mathbf{R}, \mathbf{R}, \mathbf{0})\mathbf{s}_n + [\mathbf{t}; \mathbf{0}; \mathbf{0}]$ represents the transformed source signal. The $\mathbf{s}_n$ and $\mathbf{s}_m$ denote the graph signals corresponding to points $\mathbf{x}_n$ and $\mathbf{y}_m$, respectively. The regularized Laplacian matrix is defined as $\tilde{\mathbf{L}}_m = \varepsilon\mathbf{I} + \mathbf{L}_m$, where $L_m$ denotes the graph Laplacian matrix of the local graph $G_m$ corresponding to node $m$ and $\varepsilon$ is a small positive constant to ensure numerical stability. For simplicity, we denote $\mathbf{L}_m$ as $\tilde{\mathbf{L}}_m$ hereafter. The scaling parameter $\sigma$ is automatically initialized following the strategy used in CPD (Myronenko & Song, 2010). Replacing the conventional isotropic covariance with the graph Laplacian matrix $\mathbf{L}_m$ not only retains the classical objective of minimizing the signal discrepancy between transformed source and target points, but also introduces a graph smoothness prior on the transformed signals and enforces consistency in their variation trends over the graph structure. This substitution carries stronger geometric significance, enhancing both the robustness and interpretability of the registration process. Detailed theoretical justification is provided in the Appendix.

## 3.2 Graph Signal Representation of Point Clouds

We propose a high-dimensional graph signal representation for each point as illustrated in Fig. 1(a). For a given point $\mathbf{y}_m \in \mathcal{Y}$ as an example, the corresponding graph signal is defined as $\mathbf{s}_m \in \mathbb{R}^9$, which combines three types of information: spatial coordinates $(\mathbf{y}_m^x, \mathbf{y}_m^y, \mathbf{y}_m^z)$, scaled surface normals $(\bar{\mathbf{n}}_m^x, \bar{\mathbf{n}}_m^y, \bar{\mathbf{n}}_m^z)$ and high-frequency components $(\mathbf{h}_m^1, \mathbf{h}_m^2, \mathbf{h}_m^3)$. Formally, the graph signal $\mathbf{s}_m$ is defined as

$$\mathbf{s}_m = \left(\mathbf{y}_m^x, \mathbf{y}_m^y, \mathbf{y}_m^z, \bar{\mathbf{n}}_m^x, \bar{\mathbf{n}}_m^y, \bar{\mathbf{n}}_m^z, \mathbf{h}_m^1, \mathbf{h}_m^2, \mathbf{h}_m^3\right)^T$$
$$= (\mathbf{y}_m; \bar{\mathbf{n}}_m; \mathbf{h}_m), \tag{2}$$

where $\bar{\mathbf{n}}_m \in \mathbb{R}^3$ denotes the scaled surface normal defined as $\bar{\mathbf{n}}_m = \frac{\max(\mathbf{y}) - \min(\mathbf{y})}{\max(\mathbf{n}) - \min(\mathbf{n})}\mathbf{n}_m$ ensuring scale consistency between coordinate and normal domains. Here, $\max()$ and $\min()$ compute the maximum and minimum absolute values across all matrix entries.

Next, we will describe in detail the construction process of high-frequency components. A $k$-nearest neighbor graph $\mathcal{G}_{\mathcal{Y}}$ is constructed over the point cloud $\mathcal{Y}$ based on spatial coordinates. The edge weight $\mathcal{W}_{ij}$ between two points $\mathbf{y}_i$ and $\mathbf{y}_j$ is computed by jointly considering both the Euclidean distance between coordinates and the difference in surface normals (Dinesh et al., 2022):

$$\mathcal{W}_{ij} = \exp\left(-\frac{\|\mathbf{y}_i - \mathbf{y}_j\|^2}{\sigma_p^2} - \frac{\|\mathbf{n}_i - \mathbf{n}_j\|^2}{\sigma_n^2}\right), \tag{3}$$

where the parameters $\sigma_p$ and $\sigma_n$ control the sensitivity to spatial distance and normal deviation, respectively.

In GSP, high-pass graph filters play a crucial role as they enhance the high-frequency components of signals defined on graph structures, thereby highlighting local variations and discontinuities such as edges, anomalies, and texture patterns (Ortega et al., 2018; Sandryhaila & Moura, 2013b). In our method, a high-pass graph filter is employed to amplify the differences between points while suppressing the common components shared among point cloud nodes. We adopt a linear shift-invariant FIR graph filter formulated as $\mathcal{H} = h_0\mathbf{I} + h_1\mathcal{S} + \cdots + h_n\mathcal{S}^{n-1}$ (Sandryhaila & Moura, 2013a), where $h_i > 0$ and $\mathcal{S}$ denotes the graph shift operator. In our implementation, the normalized Laplacian matrix $\mathcal{L}$ is employed as the graph shift operator to perform spectral filtering. In the Appendix, we introduced the principle of amplifying high-frequency components using high-pass graph filters. To effectively capture both geometric and structural characteristics of the point cloud, we construct three types of high-frequency components based on spatial coordinates and surface normals as follows

$$\mathbf{h}_m^1 = \|[\mathcal{L}\mathbf{y}]_{m,:}\|,$$
$$\mathbf{h}_m^2 = \|[\mathcal{L}\bar{\mathbf{n}}]_{m,:}\|, \tag{4}$$
$$\mathbf{h}_m^3 = \|\left[\left(\mathcal{L} + \mathcal{L}^2\right)\mathbf{y}\right]_{m,:}\|,$$

where $\mathbf{y}$ is the $N \times 3$ matrix composed of point coordinates $\mathbf{y}_m, \forall m = 1, ..., M$, and $\bar{\mathbf{n}}$ is defined analogously for scaled normals. The notation $[\cdot]_{m,:}$ denotes the $m$-th row of the matrix. In Theorem 1, we provide the conclusion that the high-frequency components defined in equation 4 are invariant under the rigid transformation of the point cloud, and prove it in the Appendix.

**Theorem 1** (Invariance of high-frequency components under rigid transformation). *Let* $\mathbf{x} \in \mathbb{R}^{M \times 3}$ *be the coordinate matrix of a point cloud, and* $\mathbf{y} = \mathbf{x}\mathbf{R}^T + \mathbf{T}$ *be the result of applying a rigid*

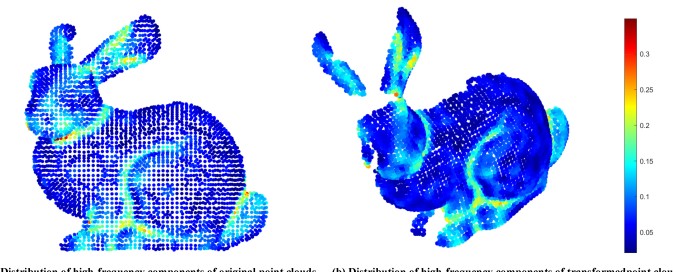

(a) Distribution of high-frequency components of original point clouds  (b) Distribution of high-frequency components of transformed point clouds

Figure 2: Visualization of high-frequency components in point cloud registration.

*transformation, where $\mathbf{R} \in \mathbb{R}^{3 \times 3}$ is a rotation matrix, and $\mathbf{T} = \mathbf{1}_M \otimes \mathbf{t}^T$ applies the translation vector $\mathbf{t} \in \mathbb{R}^3$ to each point. Then, given the graph Laplacian $\mathcal{L}$, the high-frequency components for point $\mathbf{y}_m$ defined as equation 4 are invariant under rigid transformations.*

Due to the pronounced variation of high-frequency components between different points, these components serve as effective descriptors for achieving point cloud registration. To reduce computational complexity, rather than operating on the entire dense graph matrix, we use the values of $k$ nearest neighbor indices of each node to perform weighted summation in practical calculations. To make Theorem 1 more clear and intuitive, we take the Stanford 3D Bunny as an example and visualize the energy distribution $\mathbf{h}_m^1 = ||[\mathcal{L}\mathbf{y}]_{m,:}||$ of each point, as shown in Fig. 2, where the energy is normalized. As can be seen from the Fig. 2, the high-frequency components of each point remain unchanged under the rigid transformation.

### 3.3 LOCAL GRAPH CONSTRUCTION VIA ATTRIBUTE SIMILARITY

For each node $\mathbf{y}_m$, a local graph $\mathbf{G}_m$ is constructed using its graph signal $\mathbf{s}_m$ and those of its $k$-nearest neighbors, as illustrated in Fig. 1(b). Since a single point is insufficient to characterize the underlying geodesic structure, we aggregate its $k$-nearest neighbors to jointly compute pairwise attribute similarities. Let $\mathbf{Z}_m \in \mathbb{R}^{9 \times (k+1)}$ denote the aggregated graph signals of node $\mathbf{y}_m$ and its $k$ nearest neighbors $\{\mathbf{s}_{1,m}, ..., \mathbf{s}_{k,m}\}$:

$$\mathbf{Z}_m = [\mathbf{s}_m, \mathbf{s}_{1,m}, ..., \mathbf{s}_{k,m}], \tag{5}$$

where each column corresponds to a 9-dimensional graph signal, and each row represents a specific attribute (e.g., a coordinate axis, a normal component, or a high-frequency component). Let $f_p \in \mathbb{R}^{k+1}$ denote the $p$-th row of $\mathbf{Z}_m$, corresponding to the $p$-th attribute across the local neighborhood, where $p = 1, ..., 9$. We define the adjacency matrix $\mathbf{W}_m \in \mathbb{R}^{9 \times 9}$ of the local graph $\mathbf{G}_m$ using a Gaussian kernel that measures the similarity between attribute vectors:

$$\mathbf{W}_m^{ij} = \exp\left(-\frac{\|f_p - f_q\|^2}{2\sigma_f^2}\right), \forall\, p, q = 1, ..., 9, \tag{6}$$

where $\sigma_f$ is a scalar parameter controlling the sensitivity to attribute differences. With the adjacency matrix $\mathbf{W}_m$, we compute the normalized graph Laplacian:

$$\mathbf{L}_m = \mathbf{I} - \mathbf{D}_m^{-1/2} \mathbf{W}_m \mathbf{D}_m^{-1/2}, \tag{7}$$

where $\mathbf{D}_m$ is the diagonal degree matrix. It is important to distinguish between the global graph $\mathcal{G}_{\mathcal{Y}} \in \mathbb{R}^{M \times M}$ and local graph $\mathbf{G}_m \in \mathbb{R}^{9 \times 9}$. The global graph $\mathcal{G}_{\mathcal{Y}}$ is constructed over the entire point cloud $\mathcal{Y}$ and facilitates the design of a high-pass filter for extracting high-frequency components. In contrast, the local graph $\mathbf{G}_m$ is constructed individually for each point $\mathbf{y}_m$, capturing attribute-wise relationships within its 9-dimensional graph signal $\mathbf{s}_m$.

### 3.4 EM OPTIMIZATION IN GRAPH SIGNAL REGISTRATION

Similar to probabilistic methods, we estimate the rigid transformation $\mathbf{\Theta} = \{\mathbf{R}, \mathbf{t}\}$ and the registration probabilities by maximizing the following log-likelihood function:

$$L(\mathcal{T}, \sigma^2) = -\sum_{n=1}^{N} \log p(\mathcal{T}(\mathbf{s}_n)), \tag{8}$$

Solving this optimization problem directly is computationally infeasible. To address this, the EM algorithm is employed:

**E step**: Given the previous estimate $\mathcal{T}_{\text{old}}$, Bayesian inference yields the posterior correspondence probabilities between $\mathcal{X}$ and $\mathcal{Y}$. These probabilities are organized into a matrix $\mathbf{P} \in \mathbb{R}^{M \times N}$, where each element $\mathbf{P}_{mn}$ represents the posterior probability that point $\mathbf{x}_n$ corresponds to the $m$-th component. The $\mathbf{P}_{mn}$ can be calculated as

$$\mathbf{P}_{mn} = \frac{\pi_{mn} p(\mathcal{T}_{\text{old}}(\mathbf{s}_n)|m)}{\pi_o + \sum_{m=1}^{M} \pi_{mn} p(\mathcal{T}_{\text{old}}(\mathbf{s}_n)|m)}, \tag{9}$$

where $\pi_{mn}$ denotes the prior probability that point $\mathbf{x}_n$ belongs to the $m$-th component. The $\pi_o$ is the probability density associated with outliers, whose setting follows the approach in Horaud et al. (2010). We accelerate the computation of matrix $\mathbf{P}$ through GPU parallelization and simplify the calculation of $\mathbf{P}_{mn}$ by leveraging prior probabilities to focus only on relevant indices. Furthermore, we accelerate index-based matrix rearrangement by implementing custom CUDA kernels, overcoming the inherent inefficiency of general-purpose GPU operations for such tasks.

**M step**: In the presence of outliers, the neighbor set of aligned points between the original point cloud and the target point cloud may become inconsistent, resulting in deviations in normal vector estimation and high-frequency components. Directly incorporating this information into transformation updates can introduce errors and reduce registration accuracy. To address this, in the M-step, we estimate the rigid transformation using only point coordinates, thereby mitigating the impact of noise in surface normals and high-frequency components caused by local neighborhood variations. The transformation $(\mathbf{R}, \mathbf{t})$ is computed by solving the following weighted least-squares problem:

$$Q(\mathbf{R}, \mathbf{t}, \sigma) = -\min_{\mathbf{R}, \mathbf{t}} \sum_{n=1}^{N} \sum_{m=1}^{M} \mathbf{P}_{mn} \log(\pi_{mn} p(\mathbf{x}_n|m)), \tag{10}$$

where $p(\mathbf{x}_n|m) = c_m \exp(-\frac{1}{2\sigma^2} \|\mathbf{y}_m - \mathbf{R}\mathbf{x}_n - \mathbf{t}\|^2_{\mathbf{L}_m^{\text{sub}}})$, and $\mathbf{L}_m^{\text{sub}}$ denotes the upper-left $3 \times 3$ sub-matrix of $\mathbf{L}_m$, corresponding to the spatial coordinate block. As described in Liu et al. (2021), Newton's method is employed to iteratively update the rigid transformation on the Lie group. Let $\mathbf{E}_1, \mathbf{E}_2, \ldots, \mathbf{E}_6$ represent the basis matrices of the Lie algebra. The gradient and Hessian of the objective function $Q$ are given by

$$[\nabla Q]_i = 2 \sum_{m=1}^{M} \sum_{n=1}^{N} \mathbf{P}_{mn} (\tilde{g}\widetilde{\mathbf{x}}_n - \widetilde{\mathbf{y}}_m)^T \widetilde{\mathbf{L}}_m^{\text{sub}} \tilde{g} \mathbf{E}_i \widetilde{\mathbf{x}}_n,$$

$$[\mathbf{H}]_{ij} = 2 \sum_{m=1}^{M} \sum_{n=1}^{N} \mathbf{P}_{mn} \left( \widetilde{\mathbf{x}}_n^T \mathbf{E}_j^T \tilde{g}^T \widetilde{\mathbf{L}}_m^{\text{sub}} \tilde{g} \mathbf{E}_i \widetilde{\mathbf{x}}_n + (\tilde{g}\widetilde{\mathbf{x}}_n - \widetilde{\mathbf{y}}_m)^T \widetilde{\mathbf{L}}_m^{\text{sub}} \tilde{g} \mathbf{E}_j \mathbf{E}_i \widetilde{\mathbf{x}}_n \right). \tag{11}$$

In homogeneous coordinates, we denote $\tilde{g}$, $\tilde{\mathbf{x}}_n$ and $\tilde{\mathbf{y}}_m$ as the augmented representations of rigid transformation $g$ and points $\mathbf{x}_n$, $\mathbf{y}_m$, respectively. The augmented inverse covariance matrix is constructed through direct sum operation, i.e., $\widetilde{\mathbf{L}}_m^{\text{sub}} = \mathbf{L}_m^{\text{sub}} \oplus \mathbf{0}$. Use Newton's method to update the transformation until it converges:

$$\tilde{g}_{i+1} = \tilde{g}_i \circ \exp\left( \tfrac{1}{2} (\mathbf{H} + \mathbf{H}^T)^{-1} \nabla Q \right)^{\wedge}, \tag{12}$$

where $\wedge$ denotes the mapping from a vector to its associated matrix Lie algebra element (Murray et al., 2017). After the transformation $g$ is updated, the Laplacian scaling operator $\sigma^2$ is also updated accordingly. Due to space constraints, we present only the key computational formulas here. Please refer to the Appendix for detailed derivations.

### 3.5 EFFICIENT PRIOR CONSTRUCTION FROM HIGH-FREQUENCY COMPONENT

In conventional GMM-based point cloud registration methods, the prior correspondence probability $\pi_{mn}$ between points $\mathbf{x}_n \in \mathcal{X}$ and $\mathbf{y}_m \in \mathcal{Y}$ is typically unknown. In the absence of prior knowledge, each point in $\mathcal{X}$ must be exhaustively compared with all points in $\mathcal{Y}$, resulting in high computational cost. Thanks to the invariance of high-frequency components under rigid transformation proved in Theorem 1, we propose a strategy to construct prior correspondence probabilities. We build a KD-tree using high-frequency components and retrieve the $k_{\text{match}}$ nearest neighbors in $\mathcal{Y}$ for each point in $\mathcal{X}$. The prior correspondence probability is then defined as $\pi_{mn} = 1/k_{\text{match}}$, if $\mathbf{y}_m \in \mathcal{C}_n$ and 0 otherwise, where $\mathcal{C}_n$ denotes the set of $k_{\text{match}}$ nearest candidates of $\mathbf{x}_n$ in $\mathcal{Y}$. By incorporating prior information, each point only needs to be matched with a small subset of potential candidates, thereby reducing the search space of the point cloud and greatly improving the registration efficiency.

## 4 RESULT

In this section, a series of experiments are conducted on benchmark 3D point cloud datasets to evaluate the robustness, accuracy and computational efficiency of the proposed method. All evaluations are performed in MATLAB on a workstation equipped with an NVIDIA Titan Xp GPU. The baseline methods are implemented either using code provided by the original authors or obtained from widely adopted, performance-optimized open-source libraries. The parameter settings for the baseline methods follow the authors' recommendations or software defaults when available, and are otherwise carefully tuned to ensure fairness and optimal performance. The parameter settings for our method are as follows. Similar to CPD, the initial value of $\sigma$ is automatically estimated from the data used in each experiment. In our method, the parameter was set as $\varepsilon = 0.01$. Both the global graphs ($\mathcal{G}_\mathcal{X}$, $\mathcal{G}_\mathcal{Y}$) and the local graphs ($\mathbf{G}_m$, $\forall m = 1, \ldots, M$) are constructed using $k = 10$ nearest neighbors. The parameters $\sigma_p, \sigma_n$ and $\sigma_f$ were each set to the median of their respective pairwise distances. The parameter $k_{\text{match}}$ is set to 100, 50, 50, 200, and 200 for the accuracy test, robustness test, invariance property assessment, multi-view point set, and LiDAR dataset, respectively. Following Hirose (2020), we evaluate the registration accuracy by computing the angular error (in degrees) between the estimated rotation matrix $\mathbf{R}$ and the ground-truth rotation matrix $\hat{\mathbf{R}}$, i.e.,

$$\text{AngErr}(\hat{\mathbf{R}}, \mathbf{R}) = \frac{180°}{\pi} \arccos\left(\frac{1}{2}\left(\text{Tr}(\hat{\mathbf{R}}\mathbf{R}^T) - 1\right)\right). \tag{13}$$

### 4.1 REGISTRATION ON RANGE DATASETS

*1) Accuracy test:* To evaluate the registration accuracy of the proposed method, two representative datasets were selected from the Stanford 3D Scanning Repository: Dragon and Armadillo (Curless & Levoy, 1996). To demonstrate the effectiveness of our method on large-scale point clouds, we downsample each original point cloud to approximately 20,000 points. Both datasets are captured using the Cyberware 3030MS laser scanner from two different viewpoints, with angular deviations of approximately 24° and 30°, respectively. Due to the large difference in viewing angles, the two scans only showed partial overlap, as illustrated in Fig. 3.

We compared our method with six baseline methods: GraphReg (Zhao et al., 2022), CPD (Myronenko & Song, 2010), FilterReg (point-to-point version) (Gao & Tedrake, 2019), TrICP (Chetverikov et al., 2002), LSG-CPD (Liu et al., 2021) and ECMPR (Horaud et al., 2010). CPD and FilterReg represent probabilistic method with isotropic covariance, while LSG-CPD and ECMPR represent probabilistic methods with anisotropic covariance. The TrICP serves as a robust extension of classical ICP methods, while GraphReg uses GSP to accelerate and improve the performance of physics-based point cloud registration. Due to memory limitations based on probabilistic methods, we downsampled the point clouds of ECMPR and LSG-CPD methods to approximately 8,000 points.

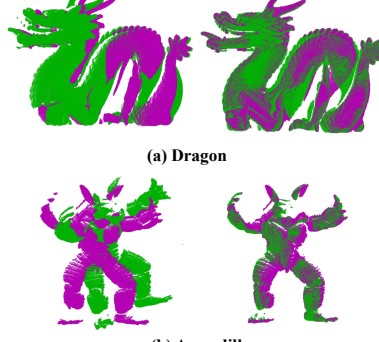

(a) Dragon

(b) Armadillo

Figure 3: Registration results of the proposed method on Dragon and Armadillo datasets.

Table 1: Registration accuracy and runtime for Dragon and Armadillo datasets.

| Method | Dragon | | Armadillo | |
|---|---|---|---|---|
| | AngErr (°) | Time (s) | AngErr (°) | Time (s) |
| **GraphReg** | 0.403 | 4.30 | 0.284 | 5.63 |
| **CPD** | 0.624 | 96.57 | 0.597 | 97.20 |
| **LSG-CPD** | 0.155 | 0.862 | 0.138 | 0.758 |
| **FilterReg** | 0.297 | 20.05 | 0.171 | 13.51 |
| **TrICP** | 0.800 | **0.50** | 0.119 | **0.55** |
| **ECMPR** | 0.203 | 133.40 | 0.173 | 137.86 |
| **GraphCPD** | **0.091** | 2.950 | **0.095** | 2.846 |

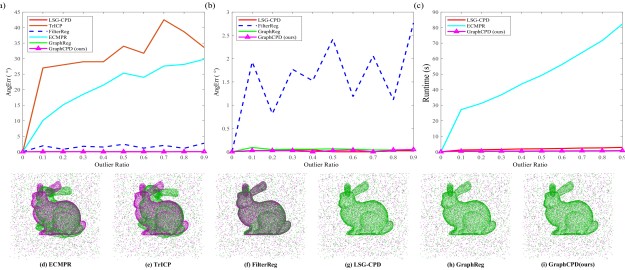

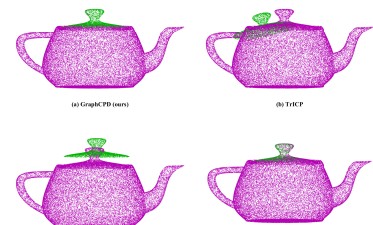

Figure 4: Outlier results. (a) Angular errors of various methods across increasing outlier ratios. (b) Zoomed-in view comparing LSG-CPD, FilterReg, GraphReg and GraphCPD for better clarity. (c) Average runtime comparison of LSG-CPD, ECMPR and GraphCPD under varying outlier ratios. (d)-(i) Visualization of registration results on data with 0.5 outliers ratio.

Figure 5: Verification of the graph invariance property on the Utah Teapot dataset.

Tab. 1 summarizes the registration accuracy and runtime. The results demonstrate that our method achieves high accuracy while maintaining efficient computation. Although other probabilistic methods such as FilterReg and ECMPR also provide relatively accurate results, their computational costs become excessively high when applied to large-scale point clouds. In contrast, ICP-based methods like TrICP and physics-based approaches like GraphReg offer faster runtimes but at the cost of reduced accuracy. In comparison, our method achieves a favorable trade-off, delivering competitive accuracy with significantly improved efficiency. Theoretically, the parameter $k_{\mathrm{match}}$ governs the construction of prior correspondence probabilities by controlling the correspondence search range. We systematically evaluated its impact, and the results in Tab. 2 show a clear trend. The increase of $k_{\mathrm{match}}$ expands the search range, thereby improving registration accuracy at the cost of longer computation time.

Table 2: Registration accuracy and runtime with different $k_{\mathrm{match}}$.

| Method | Dragon | | Armadillo | |
|---|---|---|---|---|
| | AngErr (°) | Time (s) | AngErr (°) | Time (s) |
| $K_{\mathrm{match}} = 50$ | 0.242 | 2.559 | 0.215 | 2.354 |
| $K_{\mathrm{match}} = 100$ | 0.091 | 2.950 | 0.095 | 2.846 |
| $K_{\mathrm{match}} = 150$ | 0.089 | 5.951 | 0.084 | 5.034 |
| $K_{\mathrm{match}} = 200$ | 0.074 | 7.045 | 0.078 | 7.587 |
| $K_{\mathrm{match}} = 250$ | 0.073 | 10.045 | 0.071 | 11.592 |

Table 3: Registration accuracy Comparison of Different Configurations (AngErr, °).

| Configuration | Dragon | Armadillo |
|---|---|---|
| **Laplacian Model, Weighted graph (Ours)** | **0.091** | **0.095** |
| Isotropic covariance, Weighted graph | 0.1542 | 0.3836 |
| Laplacian Model, Unweighted graph | 0.232 | 0.308 |

We conducted an ablation study to quantify the contributions of the two core components, i.e., the Graph Laplacian Model and the weighted graph-based high-pass filter. Ablations were performed by replacing these components with isotropic Gaussian covariances and an unweighted graph-based high-pass filter, respectively. As summarized in Table 3, our method achieved the best performance on both datasets. Laplacian Model outperforms isotropic covariance significantly on both datasets. The weighted graph-based high-pass filter is significantly better than the unweighted graph, demonstrating the importance of edge weighting.

*2) Robustness test:* To evaluate the robustness of the proposed method, experiments were conducted on the Bunny dataset from the Stanford 3D Scanning Repository. The original point cloud is uniformly downsampled to approximately 25,000 points, and then subjected to random rotations of around $40°$ along the $x$, $y$ and $z$ axes. To simulate varying levels of disturbance, we introduce Gaussian outliers at different ratios, defined as the number of outliers relative to the number of original points. The result in Fig. 4 shows that GraphCPD, LSG-CPD, FilterReg and GraphReg consistently achieve low angular errors across various outlier ratios, demonstrating strong robustness against outlier contamination. The performance of TrICP and ECMPR methods deteriorates rapidly as the outlier ratio increases, which are sensitive to outliers. To further evaluate the compu-

tational efficiency, a comparison is made of the runtime of several probabilistic methods in Fig. 4(c). The results indicate that GraphCPD maintains excellent robustness while ensuring low computation time.

## 4.2 Assessment of the Invariance Property

To assess the effectiveness of incorporating high-frequency components into point cloud registration, we conducted experiments on the Utah Teapot dataset (Mellado et al., 2017) and compared our method against several baseline approaches. In the experiment, the target point cloud represents the teapot, while the source point cloud corresponds to its pot cover. The objective is to accurately align the pot cover to the teapot, with the pot body treated as outliers. As illustrated in Fig. 5 and Tab. 4, our method achieves superior registration accuracy while reducing computational time. This performance improvement is mainly attributed to the richer prior information extracted from high-frequency components. In addition, high-frequency components are incorporated into the graph signal and participate in registration, effectively improving accuracy. The GraphReg method also relies on the transformation invariance of graph structure representation, thus demonstrating good estimation accuracy in this experiment. However, GraphReg adopts a one-to-one matching strategy, which makes it more susceptible to interference when facing outliers, thereby limiting its stability in high outlier scenarios.

Table 4: Registration accuracy and runtime comparison on the Utah Teapot dataset.

| Method | AngErr ($^\circ$) | Time (s) |
|---|---|---|
| TrICP | 16.739 | **0.353** |
| CPD | 0.657 | 56.84 |
| ECMPR | 39.038 | 23.678 |
| GraphReg | 0.067 | 2.344 |
| GraphCPD | **0.018** | 0.370 |

Table 5: Registration accuracy on multi-view point set.

| Method | Dragon | Happy | Armadillo |
|---|---|---|---|
| MATrICP | 2.055 | 1.903 | 2.195 |
| FilterReg | 0.405 | **0.230** | 0.177 |
| EMPMR | 0.533 | 0.235 | 0.114 |
| GraphCPD | **0.236** | 0.264 | **0.095** |

## 4.3 Evaluation on Multi-view Point Set Registration

In this subsection, we conduct experiments on three Stanford 3D Scanning Repository benchmarks: the Dragon Stand, Happy Stand, and Armadillo datasets. Each dataset comprises multiple views of a single object captured from different perspectives, along with ground-truth rigid transformations for alignment. We compare our approach against several representative baselines, including MATrICP (Li et al., 2014), EMPMR (Zhu et al., 2020) and FilterReg. Following Gao & Tedrake (2019), random perturbations are applied to the ground-truth poses, with rotation errors limited to $\pm0.05$ radians and translation errors within $\pm0.02$ meters. All point clouds are voxelized and downsampled to approximately 15,000 points per view. The average rotational error is adopted as the evaluation metric to quantify the registration accuracy.

Tab. 5 shows the average rotation error of multiple perspectives using different methods on three datasets. To more intuitively illustrate the registration performance, Fig. 6 visualizes the aligned point clouds using cross-sectional views. Experimental results show that GraphCPD achieves superior registration accuracy on two of the datasets. It is worth noting that, since our method adopts a pairwise registration strategy, error accumulation may occur along the chain of sequential transformations. Nevertheless, in comparison with existing joint multi-view registration approaches, our method still demonstrates competitive overall performance, indicating its robustness in terms of registration accuracy. As future work, we aim to explore globally consistent multi-view point cloud registration within the framework of GSP, enabling simultaneous optimization across all views and further improving overall alignment consistency and accuracy.

## 4.4 LiDAR Point Cloud Registration Evaluation

In this experiment, we evaluate our algorithm using the KITTI dataset (Geiger et al., 2012), focusing on sequence 07, which consists of a driving trajectory of nearly 700 meters. Registration pairs are constructed by sampling every two frames, yielding a total of 550 pairs for testing. To ensure computational efficiency, each point cloud is downsampled to approximately 5,000 points using a

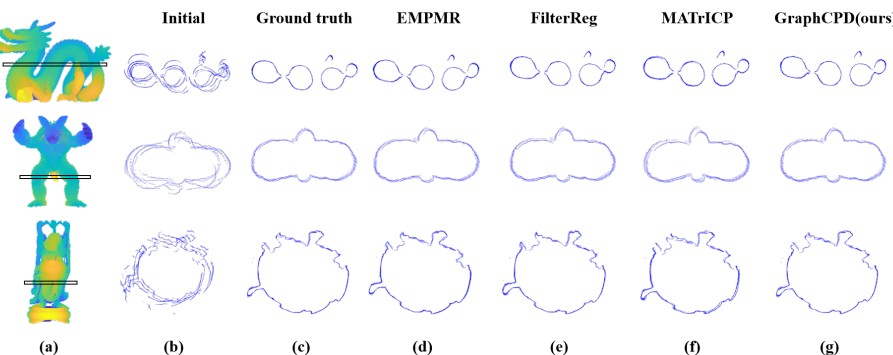

Figure 6: Multi-view registration results. The complete target model reconstructed by our method is shown, with annotated multi-view cross-sections. (b) and (c) are the initial states and corresponding ground truth. The remaining figures show the registration performance of each method in the cross-section.

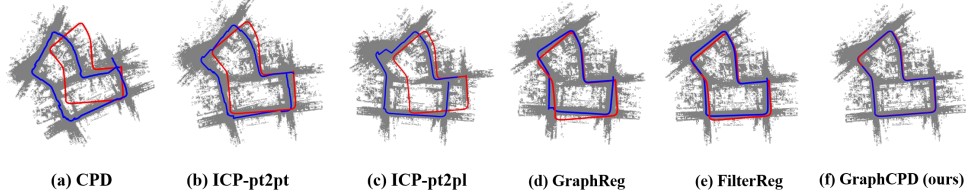

Figure 7: Qualitative evaluation of registration performance on the KITTI dataset, where red lines represent the ground truth and blue lines indicate the aligned trajectories.

Table 6: Registration performance comparison on the KITTI dataset.

| Method | Rel. Rot. (°) | Rel. Tran. (m) | Abs. Rot. (°) | Abs. Tran. (m) | Last Rot. (°) | Last Tran. (m) |
|---|---|---|---|---|---|---|
| CPD | 0.388 (25.741) | 0.345 (6.055) | 19.296 (28.999) | 46.089 (77.362) | 21.942 | 32.901 |
| ICP-pt2pt | 0.218 (5.958) | 0.289 (3.710) | 8.739 (17.833) | 16.848 (35.418) | 6.117 | 18.279 |
| ICP-pt2pl | 0.450 (17.639) | 0.178 (17.050) | 14.760 (23.590) | 44.978 (78.234) | 9.308 | 52.650 |
| GraphReg | 0.171 (11.727) | 0.183 (0.665) | 14.293 (10.838) | 6.778 (28.488) | 7.868 | 14.513 |
| FilterReg | 0.146 (3.250) | **0.074 (0.291)** | 3.548 (5.178) | 9.470 (14.187) | 3.250 | 11.999 |
| GraphCPD | **0.0292 (0.085)** | 0.108 (0.883) | **1.831 (3.581)** | **3.744 (8.738)** | **2.094** | **6.850** |

voxel grid filter. All frames are registered to the first frame to reconstruct the complete trajectory. Following the evaluation protocol in (Liu et al., 2021), we assess both relative and absolute rotation and translation errors, as well as the final deviation between the last and first frames. Translation error is defined as the Euclidean distance between the estimated and ground-truth translations, while rotation error is computed as the angular difference in the axis-angle representation. As demonstrated in Fig. 7 and Tab. 6, our proposed method exhibits superior performance in terms of rotation accuracy when compared to existing baselines.

## 5 CONCLUSION

We propose a new probabilistic registration GMM-based method, which demonstrates strong robustness, accuracy and computational efficiency. Our method utilizes a high-pass graph filter to extract high-frequency components for constructing prior probabilities, thereby improving the speed of registration. By combining high-frequency components with coordinates and normals, high-dimensional graph signals are obtained, and graph Laplacian matrices are constructed for each node to replace the traditional isotropic matrix in the probabilistic method, providing a more discriminative geometric description. In the future, we will explore globally consistent multi-view point cloud registration within the GSP framework to achieve simultaneous optimization of all views.

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

# A   APPENDIX

## A.1   THE USE OF LARGE LANGUAGE MODELS (LLMS)

We used LLMs only for light language editing (grammar, spelling, and stylistic clarity) on text written by the authors. The LLMs did not contribute to research ideation, experiment design, analysis, or writing beyond superficial edits, and it did not generate figures, tables, code, or references. All technical content was authored and verified by the authors, who take full responsibility for the manuscript. We reviewed all edits for accuracy and originality and ensured that the use of LLMs did not compromise double-blind review.

## A.2   INTERPRETABILITY OF COMPONENT-WISE DENSITY FUNCTION

This section explains the advantages of replacing the isotropic covariance in classical GMM-based methods with the Laplacian matrix $\mathbf{L}_m$. In particular, we show that $\mathbf{L}_m$ offers a more discriminative and geometrically meaningful representation of local structure, enhancing the interpretability and effectiveness of the model. The $\|\mathbf{s}_m - \mathcal{T}(\mathbf{s}_n)\|_{\hat{\mathbf{L}}_m}^2$ can be decomposed into

$$\|\mathbf{s}_m - \mathcal{T}(\mathbf{s}_n)\|_{\hat{\mathbf{L}}_m}^2 = \varepsilon \|\mathbf{s}_m - \mathcal{T}(\mathbf{s}_n)\|^2 + \|\mathbf{s}_m - \mathcal{T}(\mathbf{s}_n)\|_{\mathbf{L}_m}^2 , \tag{14}$$

where the first term on the right side of the equation is to ensure that the distance between the transformed source signal $\mathcal{T}(\mathbf{s}_n)$ and the target signal $\mathbf{s}_m$ is minimum. Expanding the second term yields:

$$\begin{aligned}
&\|\mathbf{s}_m - \mathcal{T}(\mathbf{s}_n)\|_{\mathbf{L}_m}^2 \\
=&\mathbf{s}_m{}^T \mathbf{L}_m \mathbf{s}_m + \mathcal{T}(\mathbf{s}_n)^T \mathbf{L}_m \mathcal{T}(\mathbf{s}_n) \\
&- 2 \sum_{i,j=1,..,9} \mathbf{W}_m^{ij} \left( [\mathcal{T}(\mathbf{s}_n)]_i - [\mathcal{T}(\mathbf{s}_n)]_j \right) \left( [\mathbf{s}_m]_i - [\mathbf{s}_m]_j \right),
\end{aligned} \tag{15}$$

where $\mathbf{W}_m^{ij}$ denotes the $(i,j)$-th element of the adjacency matrix $\mathbf{W}_m$ corresponding to the local graph $\mathbf{G}_m$. The term $\mathcal{T}(\mathbf{s}_n)^\top \mathbf{L}_m \mathcal{T}(\mathbf{s}_n)$ reflects the graph smoothness of the transformed source signal $\mathcal{T}(\mathbf{s}_n)$ on graph $\mathbf{G}_m$, while the cross term quantifies the consistency of local variation trends between $\mathbf{s}_m$ and $\mathcal{T}(\mathbf{s}_n)$ over the same graph. Specifically, a larger value of the cross term indicates stronger alignment in how the two signals vary across neighboring nodes. In summary, replacing the isotropic covariance with the Laplacian matrix $\mathbf{L}_m$ not only preserves the classical objective of minimizing the signal discrepancy between the transformed source and the target, but also incorporates graph characteristic that enhance the registration process. It imposes graph smoothness constraints on the transformed signal and consistency constraints on the variation trends of signals over the graph structure (Hu et al., 2021). These properties promote geometrically meaningful and transformation-consistent correspondences, leading to improved robustness and interpretability in point cloud registration.

## A.3   FREQUENCY DOMAIN INTERPRETATION OF HIGH-PASS GRAPH FILTER

Consider a linear shift-invariant graph filter (Ortega et al., 2018):

$$H = \sum_{k=0}^{K-1} h_k \mathbf{L}^k, \tag{16}$$

where $h_k > 0$, $\mathbf{L}$ is the normalized Laplacian matrix, and $K$ represents the order of the filter. Let $\mathbf{x}$ be a vertex-domain graph signal. Its Graph Fourier Transform (GFT) is (Sandryhaila & Moura, 2013b):

$$\hat{x} = \mathbf{U}^T x, \tag{17}$$

where $\mathbf{U}$ is the matrix of eigenvectors of $\mathbf{L}$, and $\hat{x}$ denotes the Fourier coefficients. The eigenvalues $\lambda_1, \ldots, \lambda_N$ of $\mathbf{L}$ form a diagonal matrix $\Lambda = \mathrm{diag}(\lambda_1, \ldots, \lambda_N)$, and are interpreted as the graph frequencies. In GSP, lower frequencies $\lambda_i$ correspond to smoother variations across the graph, i.e., the associated eigenvectors $\mathbf{u}_i$ change slowly across connected vertices. Conversely, higher frequencies correspond to more rapid variations, where the values of $\mathbf{u}_i$ differ significantly across neighboring vertices. The frequency component of $x$ at frequency $\lambda_i$ is given by:

$$\hat{x}_i = \mathbf{u}_i^T x. \tag{18}$$

For the linear shift invariant graph filter $H$, the frequency domain response is (Sandryhaila & Moura, 2013a):

$$g(\lambda) = \sum_{k=0}^{K} h_k \lambda^k. \tag{19}$$

Since $\lambda$ denotes the eigenvalues of the normalized Laplacian matrix, we have $\lambda \in (0, 2)$. Given that $h_k > 0$, $g(\lambda)$ is a monotonically increasing function over this interval. This implies that the filter amplifies high-frequency components while attenuating low-frequency components.

When the filter is applied to the signal, i.e., $y = Hx$, its GFT coefficient at frequency $\lambda_i$ becomes:

$$\hat{y}_i = g(\lambda_i)\hat{x}_i. \tag{20}$$

This shows that the output frequency component $\hat{y}_i$ is scaled by $g(\lambda_i)$. Therefore, high-frequency components (which correspond to sharp variations in the vertex domain) are enhanced, while low-frequency components (which represent smooth signal variations across the graph) are suppressed.

### A.4 INVARIANCE OF HIGH-FREQUENCY COMPONENTS UNDER RIGID TRANSFORMATION

In this section, we prove that the high-frequency components are invariant under rigid transformations in Theorem 1:

**Theorem 1** (Invariance of High-Frequency Components under Rigid Transformation) *Let $\mathbf{x} \in \mathbb{R}^{M \times 3}$ be the coordinate matrix of a point cloud, and $\mathbf{y} = \mathbf{x}\mathbf{R}^T + \mathbf{T}$ be the result of applying a rigid transformation, where $\mathbf{R} \in \mathbb{R}^{3 \times 3}$ is a rotation matrix, and $\mathbf{T} = \mathbf{1}_M \otimes \mathbf{t}^T$ applies the translation vector $\mathbf{t} \in \mathbb{R}^3$ to each point. Then, given the graph Laplacian $\mathcal{L}$, the high-frequency components for point $\mathbf{y}_m$ defined as*

$$\begin{aligned} \mathbf{h}_m^1 &= ||[\mathcal{L}\mathbf{y}]_{m,:}||, \\ \mathbf{h}_m^2 &= ||[\mathcal{L}\bar{\mathbf{n}}]_{m,:}||, \\ \mathbf{h}_m^3 &= ||\left[\left(\mathcal{L} + \mathcal{L}^2\right)\mathbf{y}\right]_{m,:}||, \end{aligned} \tag{21}$$

*are invariant under rigid transformations.*

Taking the first high-frequency component as an example:

$$\mathbf{h}_m^1 = \left\|[\mathcal{L}\mathbf{y}]_{m,:}\right\|. \tag{22}$$

Assume that integrated point cloud coordinate matrix $\mathbf{y} \in \mathbb{R}^{M \times 3}$ is generated by applying a rigid transformation to another point cloud $\mathbf{x} \in \mathbb{R}^{M \times 3}$:

$$\mathbf{y} = \mathbf{x}\mathbf{R}^T + \mathbf{T}, \quad \text{with } \mathbf{T} = \mathbf{1}_M \otimes \mathbf{t}^T.$$

Then the component can be decomposed into:

$$\begin{aligned} \mathbf{h}_m^1 &= \left\|\left[\mathcal{L}(\mathbf{x}\mathbf{R}^T + \mathbf{T})\right]_{m,:}\right\| \\ &= \left\|\left[\mathcal{L}(\mathbf{x}\mathbf{R}^T) + \mathcal{L}\mathbf{T}\right]_{m,:}\right\| \\ &= \left\|\left[\mathcal{L}\mathbf{x}\mathbf{R}^T\right]_{m,:}\right\| \\ &= \left\|[\mathcal{L}\mathbf{x}]_{m,:}\right\| = \mathbf{h}_n^1. \end{aligned} \tag{23}$$

Here, the third equality holds because $\mathbf{T}$ is constant across all rows and $\mathcal{L}\mathbf{1} = \mathbf{0}$. The fourth equality uses the orthogonality of $\mathbf{R}$, which preserves the Euclidean norm. Eq. equation 23 confirms that our high-frequency components are invariant to rigid transformations and thus well-suited for correspondence modeling in probabilistic registration frameworks. Similar derivations also apply to other high-frequency components. Thus, Theorem 1 holds true.

### A.5 DERIVATION OF EM

Recall the negative log-likelihood function in the paper. The negative log-likelihood function is:

$$
\begin{aligned}
L &= -\sum_{n=1}^{N} \log p(\mathcal{T}(\mathbf{s}_n)) \\
&= -\sum_{n=1}^{N} \log \left( \pi_o + \sum_{m=1}^{M} \pi_{mn} p(\mathcal{T}(\mathbf{s}_n)|m) \right).
\end{aligned}
\tag{24}
$$

We employ the EM algorithm to solve this problem. Within the EM framework, the correspondence between observation $\mathbf{x}_n$ and the $m$-th Gaussian mixture component is modeled as a latent variable $z_{mn} \in \{0,1\}$, where $z_{mn} = 1$ indicates that $\mathbf{x}_n$ belongs to the $m$-th component, and $z_{mn} = 0$ otherwise (Bishop & Nasrabadi, 2006). Note that $m = M + 1$ denotes an additional uniform component used to model outliers.

$$
\begin{aligned}
L_c &= - \log \prod_{n=1}^{N} (p_o^{z_{(M+1)n}} \prod_{m=1}^{M} (\pi_{mn} p(\mathcal{T}(\mathbf{s}_n)|m))^{z_{mn}}) \\
&= - \sum_{n=1}^{N} (z_{(M+1)n} \log(p_o) + \sum_{m=1}^{M} z_{mn} \log(\pi_{mn} p(\mathcal{T}(\mathbf{s}_n)|m))).
\end{aligned}
\tag{25}
$$

Focusing only on the terms involving $\mathcal{T}$ and $\sigma^2$, equation 25 simplifies to:

$$
L_c{}' = - \sum_{n=1}^{N} \sum_{m=1}^{M} z_{mn} \log(p(\mathcal{T}(\mathbf{s}_n)|m)).
\tag{26}
$$

Due to the fact that the latent variable $z_{mn}$ cannot be directly observed. Therefore, considering the transformation gold obtained in the last iteration, i.e. $E(z_{mn}|\mathcal{T}_{old}(\mathbf{s}_n))$, it is replaced by its conditional expectation. According to Bayes' rule, the posterior probability is:

$$
\begin{aligned}
&P(z_{mn} = 1|\mathcal{T}_{old}(\mathbf{s}_n)) \\
&= \frac{P(z_{mn} = 1) p(\mathcal{T}_{old}(\mathbf{s}_n)|z_{mn} = 1)}{p(\mathcal{T}_{old}(\mathbf{s}_n))} \\
&= \frac{\pi_{mn} p(\mathcal{T}_{old}(\mathbf{s}_n)|m)}{p_o + \sum_m \pi_{mn} p(\mathcal{T}_{old}(\mathbf{s}_n)|m)}.
\end{aligned}
\tag{27}
$$

Therefore, the conditional expectation is

$$
\begin{aligned}
&E(z_{mn}|\mathcal{T}_{old}(\mathbf{s}_n) \\
&= 1 \cdot P(z_{mn} = 1|\mathcal{T}_{old}(\mathbf{s}_n)) + 0 \cdot P(z_{mn} = 0|\mathcal{T}_{old}(\mathbf{s}_n)) \\
&= P(z_{mn} = 1|\mathcal{T}_{old}(\mathbf{s}_n)).
\end{aligned}
\tag{28}
$$

Define $\mathbf{P}_{mn} = P(z_{mn} = 1|\mathcal{T}_{old}(\mathbf{s}_n))$, which is (9) for E step in the paper. By replacing $z_{mn}$ in equation 26 with the conditional expectation $E(z_{mn}|\mathcal{T}_{old}(\mathbf{s}_n)$, we obtain the objective function:

$$
\begin{aligned}
\mathcal{Q} &= - \sum_{n=1}^{N} \sum_{m=1}^{M} E(z_{mn}|\mathcal{T}_{old}(\mathbf{s}_n)) \log \left( p(\mathcal{T}(\mathbf{s}_n)|m) \right) \\
&= - \sum_{n=1}^{N} \sum_{m=1}^{M} \mathbf{P}_{mn} \left( \log(c_m) - \frac{1}{2} \|\mathcal{T}(\mathbf{s}_n) - \mathbf{s}_m\|_{\mathbf{L}_m}^2 \right),
\end{aligned}
\tag{29}
$$

where

$$c_m = \left(2\pi\sigma^2\right)^{-9/2} \frac{1}{\det\left(\mathbf{L}_m^{-1}\right)^{1/2}}. \tag{30}$$

Ignoring terms independent of $\mathcal{T}$ and $\sigma^2$, the objective function becomes:

$$\mathcal{Q} = \frac{1}{2} \sum_{n=1}^{N} \sum_{m=1}^{M} \mathbf{P}_{mn} \left(9\log(\sigma^2) + \|\mathcal{T}(\mathbf{s}_n) - \mathbf{s}_m\|_{\mathbf{L}_m}^2\right). \tag{31}$$

**E step:** The corresponding probability is stored in matrix $\mathbf{P}$, where the $(m,n)$-th element $\mathbf{P}_{mn}$ represents the posterior probability of point $\mathbf{x}_n$ to be assigned to the $m$-th component:

$$\mathbf{P}_{mn} = \frac{\pi_{mn} p(\mathcal{T}_{\text{old}}(\mathbf{s}_n)|m)}{\pi_o + \sum\limits_{m} \pi_{mn} p(\mathcal{T}_{\text{old}}(\mathbf{s}_n)|m)}. \tag{32}$$

**M step:** In order to avoid matching errors in the normal vector due to different outliers, we only update the coordinate information in M step. Define 3-d rigid transformation $g \triangleq (\mathbf{R}, \mathbf{t})$. The homogeneous representations of $g$ and point $\mathbf{x}_n$ are respectively given by

$$\tilde{g} = \begin{pmatrix} \mathbf{R} & \mathbf{t} \\ \mathbf{0}^T & 1 \end{pmatrix}, \quad \tilde{\mathbf{x}}_n = \begin{pmatrix} \mathbf{x}_n \\ 1 \end{pmatrix}. \tag{33}$$

Here, group operations correspond to matrix multiplications. The transformed point under $g$ is obtained as $\tilde{g}\tilde{\mathbf{x}}_n$. The associated Lie algebra admits the following basis matrices:

$$\mathbf{E}_1 = \begin{pmatrix} 0 & 0 & 0 & 0 \\ 0 & 0 & -1 & 0 \\ 0 & 1 & 0 & 0 \\ 0 & 0 & 0 & 0 \end{pmatrix}, \mathbf{E}_2 = \begin{pmatrix} 0 & 0 & 1 & 0 \\ 0 & 0 & 0 & 0 \\ -1 & 0 & 0 & 0 \\ 0 & 0 & 0 & 0 \end{pmatrix},$$

$$\mathbf{E}_3 = \begin{pmatrix} 0 & -1 & 0 & 0 \\ 1 & 0 & 0 & 0 \\ 0 & 0 & 0 & 0 \\ 0 & 0 & 0 & 0 \end{pmatrix}, \mathbf{E}_4 = \begin{pmatrix} 0 & 0 & 0 & 1 \\ 0 & 0 & 0 & 0 \\ 0 & 0 & 0 & 0 \\ 0 & 0 & 0 & 0 \end{pmatrix}, \tag{34}$$

$$\mathbf{E}_5 = \begin{pmatrix} 0 & 0 & 0 & 0 \\ 0 & 0 & 0 & 1 \\ 0 & 0 & 0 & 0 \\ 0 & 0 & 0 & 0 \end{pmatrix}, \mathbf{E}_6 = \begin{pmatrix} 0 & 0 & 0 & 0 \\ 0 & 0 & 0 & 0 \\ 0 & 0 & 0 & 1 \\ 0 & 0 & 0 & 0 \end{pmatrix}.$$

For a function $f(\tilde{\mathbf{g}})$ defined over the Lie group, the right-trivialized derivative along the direction of a Lie algebra basis element $\mathbf{E}_i$ is given by (Chirikjian, 2011):

$$E_i^r f(\tilde{\mathbf{g}}) \triangleq \frac{d}{dt} f\left(\tilde{\mathbf{g}} \circ \exp(t\mathbf{E}_i)\right)\Big|_{t=0}. \tag{35}$$

Using the series expansion of the matrix exponential:

$$\exp(t\mathbf{E}_i) = \mathbf{I} + t\mathbf{E}_i + \frac{t^2}{2!}\mathbf{E}_i^2 + \frac{t^3}{3!}\mathbf{E}_i^3 + \cdots. \tag{36}$$

we obtain the derivative:

$$\frac{d}{dt}\exp(t\mathbf{E}_i)\Big|_{t=0} = \left(\mathbf{E}_i + t\mathbf{E}_i^2 + \frac{t^2}{2!}\mathbf{E}_i^3 + \cdots\right)\Big|_{t=0} = \mathbf{E}_i. \tag{37}$$

The gradient vector $\nabla f(\tilde{\mathbf{g}})$ and the Hessian matrix $\mathbf{H}(\tilde{\mathbf{g}})$ of the function $f(\tilde{\mathbf{g}})$ are defined using directional derivatives along the Lie algebra basis as follows:

$$\nabla f(\tilde{\mathbf{g}}) = \begin{pmatrix} E_1^r f & E_2^r f & \cdots & E_6^r f \end{pmatrix}^T,$$

$$\mathbf{H}(\tilde{\mathbf{g}}) = \begin{pmatrix} E_1^r E_1^r f & E_1^r E_2^r f & \cdots & E_1^r E_6^r f \\ E_2^r E_1^r f & E_2^r E_2^r f & \cdots & \vdots \\ \vdots & \vdots & \ddots & \vdots \\ E_6^r E_1^r f & \cdots & \cdots & E_6^r E_6^r f \end{pmatrix}. \tag{38}$$

The transformation $(\mathbf{R}, \mathbf{t})$ is obtained by solving the following weighted least-squares problem:

$$
\begin{aligned}
& Q\left(\mathbf{R}, \mathbf{t}, \sigma\right) \\
& = -\min_{\mathbf{R}, \mathbf{t}} \sum_{n=1}^{N} \sum_{m=1}^{M} \mathbf{P}_{mn} \log\left(-\frac{\pi_{mn} c_m}{2\sigma^2} \left\|\mathbf{y}_m - \mathbf{R}\mathbf{x}_n - \mathbf{t}\right\|_{\mathbf{L}_m^{\text{sub}}}^2\right).
\end{aligned}
\tag{39}
$$

By solving for the gradient and Hessian, (11) in the paper can be obtained, and the Laplacian scaling operator $\sigma^2$ is updated by

$$
\sigma^2 = \frac{\sum_{n=1}^{N} \sum_{m=1}^{M} \mathbf{P}_{mn} \left\|g(\mathbf{x}_n) - \mathbf{y}_m\right\|_{\tilde{\mathbf{L}}_m^{\text{sub}}}^2}{9 \sum_{n=1}^{N} \sum_{m=1}^{M} \mathbf{P}_{mn}}.
\tag{40}
$$

# R  RESPONSE TO REVIEWERS

We sincerely thank the reviewers for their time and constructive comments. We have carefully revised the manuscript according to the suggestions. Below is our point-by-point response.

## R1  RESPONSE TO REVIEWER 1

### R1.1  WEAKNESS 1: RESPONSE TO "CLARIFY THE DESCRIPTION IN THE MANUSCRIPT"

We sincerely thank the reviewer for carefully reading our paper and providing highly constructive comments. Regarding the many detailed issues you pointed out, we will carefully revise and improve them in the revised manuscript.

**Stronger geometric significance.** We apologize for the earlier version where the appendix was only included in the supplementary materials. Detailed derivations are provided in **Appendix A.2**. The enhanced geometric implications specifically refer to:

- Ensuring the minimized distance between the transformed source signal $\mathcal{T}(\mathbf{s}_n)$ and the target signal $\mathbf{s}_m$.
- Enforcing graph smoothness of the transformed source signal $\mathcal{T}(\mathbf{s}_n)$ on the corresponding graph $\mathbf{G}_m$.
- Imposing consistency in the local variation trends between $\mathbf{s}_m$ and $\mathcal{T}(\mathbf{s}_n)$.

**The meaning of subscripts in equation (1).** The $\mathbf{L}_m$ represents the Laplacian matrix of the local graph $\mathbf{G}_m$ corresponding to the $m$-th node in point cloud $\mathcal{Y}$. Although fidelity terms based on graph smoothing are standard practice in the field of signal restoration, traditional methods often model the entire point cloud as a single graph structure and build a global norm based on this. However, such methods often overlook the local differences between nodes when performing point cloud matching. To this end, we propose a dimensionality enhancement strategy. Specifically, we did not just focus on coordinates and normals, but instead constructed a high-dimensional graph signal for each node and based on it, created a local graph to capture the differences between points more finely.

**Comparison between GSP terminology used and GSP literature.** Traditional methods often consider multiple observed entities (such as sensors) as graph nodes, and their observed values are graph signals. In contrast, we consider the multidimensional signal on a single point cloud node $m$ as a graph signal $s_m \in \mathbb{R}^9$, with each one-dimensional attribute mapped to a virtual node. This virtual node is different from the nodes in the point cloud, but the graph signal concept used is consistent. This dimensionality enhancement processing essentially constructs a new local graph that describes the internal structure of the signal, in order to reflect the differences between different points in the same point cloud and assist in registration. Thank you very much to the reviewer for your suggestion, and we will make this clear in the manuscript.

**The parameter numerical stability $\varepsilon I$ mentioned when defining $L_m$.** The reasons for introducing the manuscript are as follows:

- The component density function can be represented as:

$$
p(\mathbf{s}_n \mid m) = c_m \exp\left(-\tfrac{1}{2\sigma^2} \left\|\mathbf{s}_m - \mathcal{T}(\mathbf{s}_n)\right\|_{L_m}^2\right) \propto \mathcal{N}\left(\mathcal{T}(\mathbf{s}_n), L_m^+\right)
$$

The covariance matrix $L_m^+$ is the pseudo inverse of the Laplacian matrix $L_m$. Since the Laplacian matrix has zero eigenvalues, its pseudo inverse is infinite in the corresponding direction, which can cause the covariance matrix to be singular and result in Gaussian distribution degradation.

- In the M step of the EM algorithm, if optimized $L_m$ directly, matrices with eigenvalues very close to zero may be generated in some iterations. This may lead to ill conditioning of the Hessian matrix, affecting convergence.

- Small regularization $\varepsilon$ can prevent algorithm overfitting.

- The introduction of term $\|\mathbf{s}_m - \mathcal{T}(\mathbf{s}_n)\|^2$ can ensure the minimum distance between the transformed source signal $\mathcal{T}(\mathbf{s}_n)$ and the target signal $\mathbf{s}_m$.

### R1.2 WEAKNESS 2: RESPONSE TO "WHETHER HIGH-FREQUENCY COMPONENTS HIGHLIGHT LOCAL VARIATIONS DEPEND ENTIRELY ON HOW THE EDGE WEIGHT ARE DEFINED."

Thank you for your insightful and thorough comments. We fully agree with your observation that unweighted graphs tend to yield higher variance in the spectral distribution compared to weighted graphs. As you rightly pointed out, when two points differ significantly while the edge weight is a small positive value, the high-frequency components do not excessively amplify such differences. Constructing a weighted graph essentially imposes adaptive smoothing constraints on the graph signal, equivalent to performing "intelligent low-pass filtering" in the graph domain.

The reason why we adopt this design is precisely to effectively reduce the variance of the spectral distribution and give it a clear geometric meaning. Point clouds often contain noise and outliers, which typically manifest as very high-frequency components. If left uncontrolled, these components can dominate the spectral distribution and obscure the mid-frequency geometric features that are more meaningful for registration. By setting an appropriate kernel bandwidth $\sigma$, we can avoid over-smoothing (which would lose geometric discriminatively) while preventing extreme high frequencies from overwhelming the mid-frequency information.

We truly appreciate your deep understanding of graph signal processing (GSP). The use of weighted graphs for extracting high-frequency information is a well-established practice in graph theory and GSP [1, 2]. To further investigate this matter, we specifically designed a comparative experiment between weighted and unweighted graphs based on Experiment 1 of the manuscript. To ensure a fair comparison, all tests were performed with a fixed iteration number of 30 and parameter $k_{match} = 100$.

Table 7: Performance comparison between weighted and unweighted graphs (AngErr, °)

| Dataset | $W_{weighted}$ | $W_{unweighted}$ |
|---|---|---|
| Dragon | 0.120 | 0.832 |
| Armadillo | 0.098 | 0.308 |

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

### R1.3 WEAKNESS 3: RESPONSE TO "GRAPH LAPLACIAN MATRIX IS CONVENTIONALLY INTERPRETED AS THE INVERSE OF THE COVARIANCE MATRIX ALSO KNOWN AS THE PRECISION MATRIX."

Thank you very much for pointing out this inaccuracy in our phrasing. Your insight into the connection with GMRFs is highly appreciated and absolutely correct. You are right that in the theories of GSP and GMRF Fields, the graph Laplacian matrix ($L$) is more precisely interpreted as the precision matrix (i.e., the inverse covariance matrix, $\Sigma^{-1}$), not the covariance matrix itself.

We have rigorously corrected this throughout the manuscript (specifically in the abstract) by changing "the graph Laplacian matrix replaces the covariance matrix" to **"the pseudo-inverse of the graph Laplacian matrix replaces the covariance matrix."**

Building upon this correction, we would like to clarify that our revised formulation is, in fact, highly consistent with the GMRF framework. In our locally constructed graph, the non-zero entries ($L_{ij} \neq 0$) in the Laplacian matrix $L$ correspond precisely to pairs of nodes $(i, j)$ that are *conditionally dependent* given all other nodes in the GMRF interpretation. This directly reflects the core "conditional independence" assumption of GM-RFs. Therefore, our use of the pseudo-inverse $L^+$ to define the local covariance structure is mathematically equivalent to defining a GMRF via a sparse precision matrix.

### R1.4 WEAKNESS 4: RESPONSE TO "THE SCALE PROBLEM OF GRAPH SIGNAL."

Thank you for this insightful comment. We acknowledge that the description of this core design in our initial manuscript was not sufficiently clear. Our response aims to clarify this issue from two perspectives:

**Rationale for Scale Normalization:** You are correct that coordinates, normals, and high-frequency components are heterogeneous by nature. This is precisely the reason for the scale normalization step we employ prior to Equation (2). Specifically, as mentioned in Section 3.2, we normalize the normal vectors as follows:

$$\bar{\mathbf{n}}_m = \frac{\max(\mathbf{y}) - \min(\mathbf{y})}{\max(\mathbf{n}) - \min(\mathbf{n})} \mathbf{n}_m$$

This step ensures that all 9 attribute dimensions are projected into a comparable numerical range. This preprocessing is a standard practice in GSP to eliminate initial scale disparities.

**Rationale of the Local-Graph Laplacian:** We wish to emphasize that for each node $\mathbf{y}_m$ in the point cloud $\mathcal{Y}$, we construct a unique, node-specific local graph $\mathcal{G}_m$ and its corresponding Laplacian matrix $\mathbf{L}_m$. In **Appendix A.2**, we explain in detail why the graph Laplace matrix has geometric explanatory significance. The term $\|\mathbf{s}_m - \mathcal{T}(\mathbf{s}_n)\|^2_{\tilde{\mathbf{L}}_m}$ can be decomposed into:

$$\|\mathbf{s}_m - \mathcal{T}(\mathbf{s}_n)\|^2_{\tilde{\mathbf{L}}_m} = \varepsilon \|\mathbf{s}_m - \mathcal{T}(\mathbf{s}_n)\|^2 + \|\mathbf{s}_m - \mathcal{T}(\mathbf{s}_n)\|^2_{\mathbf{L}_m}$$

Expanding the second term yields:

$$\begin{aligned}
&\|\mathbf{s}_m - \mathcal{T}(\mathbf{s}_n)\|^2_{\mathbf{L}_m} \\
=& \mathbf{s}_m{}^T \mathbf{L}_m \mathbf{s}_m + \mathcal{T}(\mathbf{s}_n)^T \mathbf{L}_m \mathcal{T}(\mathbf{s}_n) \\
&- 2 \sum_{i,j=1,..,9} \mathbf{W}_m^{ij} \left( [\mathcal{T}(\mathbf{s}_n)]_i - [\mathcal{T}(\mathbf{s}_n)]_j \right) \left( [\mathbf{s}_m]_i - [\mathbf{s}_m]_j \right)
\end{aligned}$$

The term $\mathcal{T}(\mathbf{s}_n)^\top \mathbf{L}_m \mathcal{T}(\mathbf{s}_n)$ reflects the graph smoothness of the transformed source signal. The cross term quantifies the consistency of local variation trends. In summary, replacing the isotropic covariance with the Laplacian matrix $\mathbf{L}_m$ incorporates graph characteristics that enhance the registration process by imposing graph smoothness and consistency constraints.

### R1.5 WEAKNESS 5: RESPONSE TO "TABLE 1 EXPERIMENTAL RESULTS AND PERFORMANCE ANALYSIS."

We sincerely thank the reviewer for this important viewpoint. In the updated code (https://anonymous.4open.science/r/GraphCPD-801E), we have optimized the calculations to improve competitiveness:

- **GPU Parallel Computing:** We reformulated the E-step computations as parallelizable matrix operations on GPU.

- **Neighborhood-based Probability Simplification:** We utilized $k_{match}$ nearest neighbors of the highest frequency components to simplify probability calculations.

- **CUDA Kernel Optimization:** We implemented custom CUDA kernels to address matrix rearrangement bottlenecks.

**In the revised manuscript**, we re-evaluated LSG-CPD and ECMPR by downsampling to 8000 nodes to fairly compare accuracy and runtime. The updated results are presented below:

### R1.6 WEAKNESS 6: RESPONSE TO "THE APPENDIX."

Thank you for this meticulous reminder. We have now integrated all relevant content from the supplementary material into the main manuscript as a formal appendix section.

Table 8: Registration accuracy and runtime for Dragon and Armadillo datasets.

| Method | Dragon AngErr (°) | Dragon Time (s) | Armadillo AngErr (°) | Armadillo Time (s) |
|---|---|---|---|---|
| GraphReg | 0.403 | 4.30 | 0.284 | 5.63 |
| CPD | 0.624 | 96.57 | 0.597 | 97.20 |
| LSG-CPD | 0.155 | 0.862 | 0.138 | 0.758 |
| FilterReg | 0.297 | 20.05 | 0.171 | 13.51 |
| TrICP | 0.800 | 0.50 | 0.119 | 0.55 |
| ECMPR | 0.203 | 133.40 | 0.173 | 137.86 |
| GraphCPD | **0.091** | 2.950 | **0.095** | 2.846 |

## R2 RESPONSE TO REVIEWER 2

### R2.1 WEAKNESS 1: RESPONSE TO "NOVELTY".

We believe the main innovation lies in fundamentally innovating the classic CPD framework from a new perspective of graph signal processing. We address two core flaws: the isotropic covariance problem and the inefficient uniform prior search. Our **key innovations** are:

- **Representation Dimensionality Expansion:** We move beyond points to "Graph Signals" with local graph modeling. We construct a high-dimensional descriptor graph for each point.

- **Structured Covariance:** We employ the pseudo-inverse of the Graph Laplacian as structured covariance to incorporate geometric meaning ($L_m^+$ instead of $\sigma^2 I$).

- **Efficient Priors:** We construct efficient priors through high-frequency invariance, reducing computational complexity by an order of magnitude, guaranteed by Theorem 1.

Regarding GraphReg, it combines hard matching with physical models, whereas GraphCPD combines soft matching with probabilistic models. Our soft allocation is more robust to outliers and incorrect matches compared to the hard matching in GraphReg.

### R2.2 WEAKNESS 2: RESPONSE TO "COMPARISONS WITH CLASSICAL SPECTRAL DESCRIPTORS (HKS/WKS)."

HKS and WKS essentially perform low-pass filtering, smoothing out high-frequency details. This contradicts our intention of obtaining high-frequency components. Although we can construct high pass filters based on variants of HKS/WKS, for example taking $g_{high}(L) = I - \beta \exp(-tL)$, the resulting matrix loses sparsity. Consider a sample Laplacian $L$:

$$L = \begin{bmatrix} 1.0 & -0.3 & 0.0 & 0.0 & -0.7 & 0.0 \\ -0.3 & 1.0 & 0.0 & -0.1 & -0.6 & 0.0 \\ 0.0 & 0.0 & 1.0 & -0.5 & 0.0 & -0.5 \\ 0.0 & -0.1 & 0.0 & 1.0 & 0.0 & -0.9 \\ -0.7 & -0.3 & 0.0 & 0.0 & 1.0 & 0.0 \\ 0.0 & -0.1 & -0.9 & 0.0 & 0.0 & 1.0 \end{bmatrix}$$

The corresponding high-pass filter $g_{high}(L)$ becomes dense (non-sparse). Our method maintains sparsity (high computational efficiency), provides significant high-frequency enhancement, and has wide applicability.

### R2.3 WEAKNESS 3: RESPONSE TO "DISCRIMINATIVE POWER OF THE DESCRIPTOR."

We have supplemented the manuscript with a visualization analysis (Fig. 2) using the Bunny dataset. It clearly observes that although the point cloud undergoes rigid transformations, the distribution pattern of its high-frequency components remains consistent, visually validating **Theorem 1**.

### R2.4 WEAKNESS 4: RESPONSE TO "BASELINE SELECTION."

Deep learning methods primarily focus on **coarse registration**, whereas our work and the selected baselines focus on **fine registration**. As shown in the table below (using KITTI dataset results), our method achieves higher precision than current deep learning methods in this specific task.

Table 9: Comparison with Deep Learning Methods (KITTI)

| Methods | RTE (cm) | RRE (°) |
|---------|----------|---------|
| FCGF [2] | 4.881 | 0.17 |
| CoFiNet [3] | 8.5 | 0.41 |
| GeoTransformer [4] | 7.4 | 0.27 |
| DCATr [5] | 6.6 | 0.22 |
| PARE-Net [6] | 4.9 | 0.23 |
| CAST [7] | 2.5 | 0.27 |
| ML-SemReg [8] | 5.2 | 0.2 |
| **GraphCPD (ours)** | **0.108** | **0.0292** |

### R2.5 WEAKNESS 5: RESPONSE TO "EFFICIENCY OPTIMIZATION."

We have optimized our code via GPU Parallel Computing, Neighborhood-based Probability Simplification, and CUDA Kernel Optimization. For the simple Utah Teapot dataset, processing time is now **0.37s**. Code: `https://anonymous.4open.science/r/GraphCPD-801E`.

### R3 RESPONSE TO REVIEWER 3

### R3.1 WEAKNESS 1: RESPONSE TO "SELECTION OF DATASET."

We have conducted additional experiments on the ETH terrestrial laser scanning (TLS) dataset. The results confirm our method maintains superior registration accuracy.

Table 10: Registration accuracy on the ETH Facade dataset.

| | CPD | ICP-pt2pt | ICP-pt2pl | GraphReg | FilterReg | GraphCPD |
|---|-----|-----------|-----------|----------|-----------|----------|
| Pair 1 | 9.024 | 7.254 | 8.246 | 1.952 | 0.572 | **0.168** |
| Pair 2 | 5.221 | 3.254 | 5.781 | 2.874 | 1.233 | **0.559** |
| Pair 3 | 0.722 | 5.247 | 6.254 | 5.751 | 0.932 | **0.166** |
| Pair 4 | 1.931 | 3.254 | 1.254 | 3.922 | 0.528 | **0.050** |
| Pair 5 | 8.113 | 6.987 | 5.249 | 3.728 | 1.95 | **0.277** |

### R3.2 WEAKNESS 2: RESPONSE TO "CPD AND LSG-CPD METHODS."

We have updated Tables 1-3 to include CPD results. Regarding LSG-CPD in multi-view and KITTI experiments: LSG-CPD has a complexity of $O(MN)$, whereas ours is $O(kN)$. We excluded it from large-scale accuracy-only tests due to this efficiency gap, but demonstrated in Experiment 1 that we achieve comparable accuracy with much higher speed.

### R3.3 WEAKNESS 3: RESPONSE TO "COMPARISON OF PARAMETER $k_{match}$."

We have provided a clearer explanation of $k_{match}$ in the main text. The table below shows the trade-off: larger $k_{match}$ improves accuracy but increases time.

Table 11: Effect of $k_{match}$ on Dragon and Armadillo datasets.

| Method | Dragon AngErr (°) | Dragon Time (s) | Armadillo AngErr (°) | Armadillo Time (s) |
|--------|-------------------|-----------------|----------------------|--------------------|
| GraphCPD($K_{match} = 50$) | 0.242 | 2.559 | 0.215 | 2.354 |
| GraphCPD($K_{match} = 100$) | 0.091 | 2.950 | 0.095 | 2.846 |
| GraphCPD($K_{match} = 150$) | 0.089 | 5.951 | 0.084 | 5.034 |
| GraphCPD($K_{match} = 200$) | 0.074 | 7.045 | 0.078 | 7.587 |
| GraphCPD($K_{match} = 250$) | 0.073 | 10.045 | 0.071 | 11.592 |

### R3.4 WEAKNESS 4: RESPONSE TO "ABLATION STUDIES".

We conducted ablation studies comparing Weighted vs. Unweighted Graph Laplacian and Isotropic Covariance.

Table 12: Performance Comparison of Different Configurations (AngErr, °)

| Configuration | Dragon | Armadillo |
|---|---|---|
| **Graph Laplacian, Weighted (Ours)** | **0.091** | **0.095** |
| Isotropic, Weighted | 0.1542 | 0.3836 |
| Graph Laplacian, Unweighted | 0.232 | 0.308 |

### R3.5 QUESTION 1: RESPONSE TO "EXPLANATION OF TIME CONSUMPTION FOR LSG-CPD IN TABLE 1."

LSG-CPD requires computing matching probabilities for all point pairs ($O(MN)$). With ~20,000 points, the computation grows by a factor of ~32.6x per iteration compared to the original baseline of 3,500 points. To address this in the revised manuscript, we downsampled point clouds to 8,000 points for LSG-CPD testing to ensure fair and practical runtime comparison.

