# OpenReview forum: "GraphCPD: Coherent Point Drift for Point Cloud Registration via Graph Signal Processing"
_ICLR.cc/2026/Conference — Submitted to ICLR 2026_

### Official Review · Reviewer_VvGA · 2025-10-31

**Soundness:** 3
**Presentation:** 2
**Contribution:** 3
**Rating:** 4
**Confidence:** 5

**Summary:**

This paper proposes GraphCPD, a probabilistic point cloud registration method integrating Graph Signal Processing (GSP) with GMM. The method uses high-pass graph filters to extract transformation-invariant high-frequency components, replaces GMM's isotropic covariance with point-specific graph Laplacian matrices, and constructs 9-D graph signals from coordinates, normals, and high-frequency components.

**Strengths:**

1. Novel integration of GSP into probabilistic registration with well-motivated use of local Laplacians for capturing geometry.

2. Theorem 1 proofs the transformation invariance of high-frequency components.

3. Promising accuracy/efficiency trade-offs on some benchmarks (Table 1, Utah Teapot).

**Weaknesses:**

1. Experimental scope is narrow, using limited datasets (Stanford scans, Utah Teapot, KITTI) without broader evaluation on other standard benchmarks.

2. Missing critical baselines. CPD is absent from Tables 1-3 (only in Table 4), weakening comparisons to CPD-style methods. LSG‑CPD is omitted from Sec. 4.3 (multi‑view) and Sec. 4.4 (KITTI) despite using similar experimental configurations.

3. The $k_{match}$ claim lacks substantiation. The paper attributes lower accuracy vs. LSG-CPD to "small $k_{match}$" but doesn't state the value in main text (Maybe $k_{match}=100$?). Providing AngErr and Time results with different $k_{match}$ values (such as 100, 200, 500) in Table 1 would be helpful.


4. No ablation studies to validate individual components (such as Laplacian $\|\cdot\|_{\mathcal{L}_m}$. vs. isotropic).

**Questions:**

- Why LSG-CPD spends so much time in Table 1? Since it claims 89ms for 3500 points in their paper.

- Why the results of LSG-CPD are absent in Table 3 and Table 4?

---

> ### Author Response · Authors · 2025-11-19
> **Response to "3.2. CPD and LSG-CPD methods."**
>
> We sincerely appreciate the reviewer's insightful comments and have addressed them point by point below.
>
> 1.The CPD method was missing in Tables 1-3. We thank the reviewer for pointing this out and have now updated the results of CPD in Tables 1 to 3.
>
>
>
> Table 1: Registration accuracy and runtime for Dragon and Armadillo datasets.
>
> | Method    | Dragon AngErr (°) | Dragon Time (s) | Armadillo AngErr (°) | Armadillo Time (s) |
> | --------- | ----------------- | --------------- | -------------------- | ------------------ |
> | GraphReg  | 0.403             | 4.30            | 0.284                | 5.63               |
> | CPD       | 0.624             | 96.57           | 0.597                | 97.20              |
> | LSG-CPD   | 0.155             | 0.862           | 0.138                | 0.758              |
> | FilterReg | 0.297             | 20.05           | 0.171                | 13.51              |
> | TrICP     | 0.800             | 0.50            | 0.119                | 0.55               |
> | ECMPR     | 0.203             | 133.40          | 0.173                | 137.86             |
> | GraphCPD  | **0.091**         | 2.950          | **0.095**            | 2.846            |
>
> Table 2: Registration accuracy and runtime comparison on the Utah Teapot dataset.
>
> | Method       | AngErr (°) | Time (s)  |
> | ------------ | ---------- | --------- |
> | **TrICP**    | 16.739     | **0.353** |
> | **CPD**      | 0.657      | 56.84     |
> | **ECMPR**    | 39.038     | 23.678    |
> | **GraphReg** | 0.067      | 2.344     |
> | **GraphCPD** | **0.018**  | 0.371     |
>
>
> **Note**:   In this modification, we also optimized the experimental speed of the code by using GPU acceleration, utilizing sparse computing, and utilizing CUDA scripts. Please refer to the response to reviewer SbMu for specific details (Response to "2.5. Efficiency optimization"). New code link: https://anonymous.4open.science/r/GraphCPD-801E.
>
> 2.Regarding the absence of LSG-CPD in the multi-view and KITTI experiments. We appreciate the reviewer's question. The reason LSG-CPD was not included in these experiments is that they were designed to evaluate only registration accuracy (following the evaluation protocols used in the GraphReg and LSG-CPD papers). Our method aims to address the low efficiency of conventional GMM-based approaches. To achieve this, each point in our method is matched with only $k_{match}$  nodes, which significantly improves speed with only minor accuracy loss. In contrast, LSG-CPD still adopts the traditional policy of matching every node based on the GMM approach. As a result, our approach has a computational complexity of O(kN), while LSG-CPD has a complexity of O (MN), with $k \ll M$. Therefore, we did not include LSG-CPD in experiments focused solely on accuracy evaluation.

---

> ### Author Response · Authors · 2025-11-19
> **Response to "3.3. Comparison of parameter $k_{match}$."**
>
> We fully understand the reviewer's confusion regarding the parameter settings in our previous manuscript. Due to space constraints in the main text, we had placed the detailed explanation of $k_{match}$ settings in Section A.6 "EXPERIMENT DETAILS" of the appendix, with a note in the experimental section in the  stating: "The specific parameter setting will be explained in the Appendix."
> Additionally, we included a comparative analysis of different $k_{match}$ parameter settings in the appendix, and mentioned in the main text that: "The larger the $k_{match}$, the higher the accuracy. This will be demonstrated experimentally in the Appendix."
> We sincerely apologize for any misunderstanding this may have caused. In response, we have now provided a clearer explanation of the parameter settings directly within the experimental section of the main text. We greatly appreciate the reviewer's suggestion to include more comprehensive results. Accordingly, we have supplemented Table 1 with AngErr and computational time results corresponding to different $k_{match}$ values.
>
> | Method                        | Dragon AngErr (°) | Dragon    Time (s) | Armadillo AngErr (°) | Armadillo Time (s) |
> | ----------------------------- | ----------------- | ------------------ | -------------------- | ------------------ |
> | **GraphCPD($K_{match}=50$)**  | 0.242             | 2.559              | 0.215                | 2.354              |
> | **GraphCPD($K_{match}=100$)** | 0.091             | 2.950              | 0.095                | 2.846              |
> | **GraphCPD($K_{match}=150$)** | 0.089             | 5.951              | 0.084                | 5.034              |
> | **GraphCPD($K_{match}=200$)** | 0.074             | 7.045              | 0.078                | 7.587              |
> | **GraphCPD($K_{match}=250$)** | 0.073             | 10.045             | 0.071                | 11.592             |
>
> As discussed in our paper, $k_{\mathrm{match}}$ controls the construction of prior correspondence probabilities. These findings validate our hypothesis that as $k_{\text{match}}$ increases, the corresponding search range widens, leading to improved registration accuracy, though at the cost of increased computational time. This trade-off perfectly illustrates our claim that "our method achieves a favorable trade-off, delivering competitive accuracy with significantly improved efficiency."

---

> ### Author Response · Authors · 2025-11-19
> **Response to "3.4. Ablation studies".**
>
> We sincerely thank the reviewer for their valuable reminder. The graph Laplacian matrix is indeed a core innovation of our paper, and we agree that corresponding ablation studies are essential. In response to this suggestion, we have supplemented the experimental comparisons with isotropic covariance based on **Experiment 1** in the manuscript. Additionally, we have conducted ablation studies on other components, specifically comparing the performance using weighted graphs versus unweighted graphs. The experimental results are summarized in the table below
>
> ### Table 1. Performance Comparison of Different Configurations  (AngErr, °)
>
> | Configuration                        | Dragon    | Armadillo |
> | ------------------------------------ | --------- | --------- |
> | **Graph Laplacian, Weighted (Ours)** | **0.091** | **0.095** |
> | Isotropic, Weighted                  | 0.1542    | 0.3836    |
> | Graph Laplacian, Unweighted          | 0.232     | 0.308     |
>
> From the results, it can be seen that our proposed method (weighted Graph Laplacian) achieved the best performance on both datasets. Weighted Graph Laplacian outperforms isotropic covariance significantly on both datasets. The weighted graph is significantly better than the unweighted variant, demonstrating the importance of edge weighting.  We will integrate the above results into the main manuscript.

---

> ### Author Response · Authors · 2025-11-19
> **Response to "Question 3.1. Explanation of Time Consumption for LSG-CPD in Table 1."**
>
> We sincerely thank the reviewer for raising this important question regarding the computational time of LSG-CPD reported in Table 1. We appreciate the opportunity to clarify this point.
>
> 1. LSG-CPD is built upon the classical CPD framework, which requires computing the matching probabilities between all point pairs in the source and target point clouds, resulting in a computational complexity of O(MN), where M and N represent the nodes of point cloud $\mathcal{Y}$ and  $\mathcal{X}$,  respectively. This implies that as the number of points increases, the computation time grows quadratically. In the original LSG-CPD paper, the reported processing time for 3,500 points was 89 ms, representing baseline performance on relatively small-scale point clouds. In our experiments, the point cloud scale reached approximately 20,000 points. Consequently, the number of matching operations per iteration increased from 3,500² (≈ 12.25 million) to 20,000² (≈ 400 million), representing a growth factor of approximately **32.6 times per iteration**.  Furthermore, point cloud registration is an iterative process, and LSG-CPD typically requires **50 iterations** to converge (as configured in the LSG-CPD paper). Each iteration necessitates recalculating all point pair matching probabilities: a single iteration would take approximately 2.9s(based on the above estimation). With 50 iterations,  the total theory time will increase to 145s.
> 2. We fully understand the reviewer's concern. In response to the reviewer’s valid concern about runtime efficiency, we have taken steps to optimize the experimental process. We downsampled the point clouds to 8,000 points when testing the LSG-CPD method in the revised manuscript.  This adjustment led to a significant reduction in runtime, with a marginal decrease in accuracy. The updated results will be clearly reflected in the revised table and discussed accordingly in the text. The following table is the results of our update, and corresponding revisions will be made in the updated manuscript.
>
> Table 1: Registration accuracy and runtime for Dragon and Armadillo datasets.
>
> | Method    | Dragon AngErr (°) | Dragon Time (s) | Armadillo AngErr (°) | Armadillo Time (s) |
> | --------- | ----------------- | --------------- | -------------------- | ------------------ |
> | GraphReg  | 0.403             | 4.30            | 0.284                | 5.63               |
> | CPD       | 0.624             | 96.57           | 0.597                | 97.20              |
> | LSG-CPD   | 0.155             | 0.862           | 0.138                | 0.758              |
> | FilterReg | 0.297             | 20.05           | 0.171                | 13.51              |
> | TrICP     | 0.800             | 0.50            | 0.119                | 0.55               |
> | ECMPR     | 0.203             | 133.40          | 0.173                | 137.86             |
> | GraphCPD  | **0.091**         | 2.950           | **0.095**            | 2.846              |

---

> ### Author Response · Authors · 2025-11-20
> **Response to "3.1. Selection of dataset."**
>
> We sincerely thank the reviewer for this insightful comment.  We have conducted additional experiments on the ETH terrestrial laser scanning (TLS) dataset [1], a standard and challenging benchmark in the field. This dataset features large-scale static scenes with significant viewpoint changes, providing a complementary evaluation scenario to our existing experiments. The new results  show that our method, GraphCPD, maintains superior registration accuracy on this additional benchmark. This strengthens our claim regarding the robustness and general applicability of our approach across diverse data types and real-world challenges.    Our experimental results are as follows:
>
>
>
> We assess our method on the ETH terrestrial laser scanning (TLS) dataset [1], a standard benchmark in diverse fields like geomatics, manufacturing and medicine.  Using five point cloud pairs from the Facade dataset, we downsampled each point cloud to approximately 20,000 points via voxel grid filter.  The Tab. 1 provides the registration accuracy results. The results confirm that our method outperforms existing methods in registration accuracy.
>
>  Table 1. Registration accuracy on the ETH Facade dataset.
>
> |        | CPD   | ICP-pt2pt | ICP-pt2pl | GraphReg | FilterReg | GraphCPD  |
> | ------ | ----- | --------- | --------- | -------- | --------- | --------- |
> | Pair 1 | 9.024 | 7.254     | 8.246     | 1.952    | 0.572     | **0.168** |
> | Pair 2 | 5.221 | 3.254     | 5.781     | 2.874    | 1.233     | **0.559** |
> | Pair 3 | 0.722 | 5.247     | 6.254     | 5.751    | 0.932     | **0.166** |
> | Pair 4 | 1.931 | 3.254     | 1.254     | 3.922    | 0.528     | **0.050** |
> | Pair 5 | 8.113 | 6.987     | 5.249     | 3.728    | 1.95      | **0.277** |
>
> [1] Terrestrial Laser Scanner Point Clouds. [Online]. Available,  [Automatic registration of partially overlapping terrestrial laser scanner point clouds – Photogrammetry and Remote Sensing | ETH Zurich](https://prs.igp.ethz.ch/research/completed_projects/automatic_registration_of_point_clouds.html).

---

### Official Review · Reviewer_SbMu · 2025-11-01

**Soundness:** 2
**Presentation:** 2
**Contribution:** 2
**Rating:** 2
**Confidence:** 4

**Summary:**

The paper introduces GraphCPD, a probabilistic point-set registration framework that integrates ideas from graph signal processing (GSP). The authors construct a local high-frequency descriptor by applying a graph Laplacian filter to point coordinates and normals, claiming that these descriptors are rigid-motion invariant and can guide correspondence estimation. The method replaces the isotropic Gaussian covariance in classical CPD with a local Laplacian-based metric and uses high-frequency responses to constrain candidate matches. Experimental results on several classical rigid-registration datasets show improved accuracy over traditional CPD variants.

**Strengths:**

- Mathematically sound integration of GSP principles into a probabilistic registration framework (CPD).
- The method is interpretable and relatively general, it could potentially be extended to non-rigid or semantic alignment.

**Weaknesses:**

- Limited novelty. The use of graph-Laplacian high-frequency responses as local geometric descriptors is conceptually not novel, having been explored in prior graph/spectral registration work (e.g., GraphReg and classical spectral descriptors). The paper mainly repackages a hand-crafted high-pass descriptor within a standard CPD framework, which is an incremental recombination rather than a new principle.

- Missing comparisons with classical spectral descriptors (HKS/WKS). The proposed high-pass descriptor is positioned as a GSP-based local feature, yet the paper does not compare against Heat Kernel Signature (HKS) or Wave Kernel Signature (WKS), canonical spectral descriptors derived from the Laplace(-Beltrami) operator that can be viewed as graph-spectral filters on discrete meshes/point clouds. These descriptors are rigid-motion invariant (and WKS is relatively more scale-stable), and thus constitute strong, conceptually proximate baselines. It's unclear whether the proposed high-pass design is more distinctive or robust than standard spectral alternatives.

- Questionable discriminative power of the descriptor. Although the authors claim the high-pass filtering highlights salient local geometry, no visualization or empirical analysis (e.g., t-SNE embedding of descriptors across corresponding/non-corresponding points) is provided to verify its discriminative ability. Without such evidence, it is hard to judge whether the descriptor truly helps correspondence estimation or merely serves as a heuristic prior.

- Outdated baseline selection and limited comparison. The evaluation mainly compares with older non-learning methods (e.g., CPD, ECMPR, FilterReg), while no comparison to modern deep-learning descriptors (such as Predator and GeoTransformer) is provided. This limits the relevance of the reported improvements in the current research landscape.

- Efficiency concerns. The claimed “fast” method still requires over 4 seconds even for the simple Utah Teapot model, which is quite high for small-scale rigid alignment. Given the algorithm’s largely analytical nature and lack of GPU acceleration, the practicality and scalability are questionable.

**Questions:**

- Have you compared the proposed high-frequency descriptors with diffusion-based ones (e.g., HKS, WKS) in terms of invariance and discriminability?
- Could you provide a feature-space visualization (e.g., t-SNE) showing correspondence similarity for your descriptors?
- Would integrating a learned feature extractor outperform the hand-crafted descriptor? The authors should compare more learning-based deep geometric descriptors, such as Predator and GeoTransformer, to confirm the SOTA performance.

---

> ### Author Response · Authors · 2025-11-18
> **Response to "2.5. Efficiency optimization".**
>
> We sincerely thank you for your valuable comments and fully understand your concern. Based on your feedback, we have thoroughly optimized our code with the following three major improvements:
>
> ## 1. GPU Parallel Computing
>
> To enhance the efficiency of our method, we have reformulated the computations in the E-step of the EM algorithm as parallelizable matrix operations that can be efficiently executed on GPU hardware.
>
> ## 2. Neighborhood-based Probability Simplification
>
> We simplified the probability calculation for $P$ in the E-step by leveraging the $k\_{match}$ nearest neighbors of the highest frequency components. Specifically, for the $m$-th node, we only compute probabilities $p\_{mn}$ for nodes where $\mathbf{y}\_m \in \mathcal{C}\_n$, where $\mathcal{C}\_n$ represents the set of $k\_{match}$ closest candidates to the $m$-th node.
>
> Furthermore, we optimized the computation of the term $\left( { - \tfrac{1}{{2{\sigma ^2}}}\left\| \|{{{\mathbf{s}}\_m} -\mathcal{T}({{\mathbf{s}}\_n})} \right\|\ |\ _{{{\tilde {\mathbf{L}}}}\_m}^2} \right)$ in $P\_{mn}$.
>
> ## 3. CUDA Kernel Optimization
>
> To further accelerate performance, we implemented custom CUDA kernels. Since GPUs are not inherently efficient at matrix rearrangement based on indices, our custom CUDA scripts specifically address this bottleneck.
>
> We are grateful for your suggestions, which have led to significant speed improvements. **For the Utah Teapot dataset, we have achieved a processing time of 0.37s. We retested  the time for all our experiments and will revise it in a new manuscript. We believe these enhancements make our method more competitive.** The updated code has been made available at the anonymous link for the reviewers' reference. https://anonymous.4open.science/r/GraphCPD-801E.

---

> ### Author Response · Authors · 2025-11-18
> **Response to "2.4. Baseline selection."**
>
> Thank you for your valuable suggestions. Your point about the need to compare baseline selections with modern deep learning methods is very insightful, and this indeed would provide a more comprehensive evaluation of the paper. We have given this careful consideration and would like to clarify our position in our response.
>
> As pointed out in the survey [1], "*It is interesting that deep learning methods are rare in this area, potentially due to the difficulties in ultra-accurate error prediction.*" Currently,  deep learning methods primarily focus on addressing **coarse registration** problems to provide a good initial transformation.
>
> In contrast, our work (along with the baseline methods we selected) belongs to the category of **fine registration**, with the goal of achieving extremely high final alignment accuracy. In this core task, traditional non-learning methods (such as ICP and GMM) are still regarded as the standard paradigm and remain dominant due to their robustness in minimizing residual errors.
>
> Therefore, directly comparing deep learning methods with our approach involves inherently different task objectives and evaluation criteria. Additionally, the robustness of deep learning methods in challenging scenarios such as low overlap rates and extreme noise remains to be fully tested, and their generalization capability is sensitive to training data distribution. In comparison, non-learning methods often demonstrate greater stability in these aspects. The following table shows the accuracy results of several deep learning methods on the KITTI dataset, demonstrating that the precision of deep learning methods is lower than ours:
>
> | Methods             | RTE (cm)  | RRE (°)    |
> | ------------------- | --------- | ---------- |
> | FCGF [2]            | 4.881     | 0.17       |
> | CoFiNet [3]         | 8.5       | 0.41       |
> | GeoTransformer [4]  | 7.4       | 0.27       |
> | DCATr [5]           | 6.6       | 0.22       |
> | PARE-Net [6]        | 4.9       | 0.23       |
> | CAST [7]            | 2.5       | 0.27       |
> | ML-SemReg [8]       | 5.2       | 0.2        |
> | **GraphCPD (ours)** | **1.08** | **0.03** |
>
> We would like to express our gratitude once again to the reviewer for their valuable insights. As we have responded in the revised manuscript, we have provided a clearer explanation of the core task boundary between deep learning methods (focusing on coarse registration) and our work (focusing on fine registration) in the **related work in the manuscript**.
>
> ## References
>
> [1] Jiaqi Yang, Chu'ai Zhang, Zhengbao Wang, et al.  *3D registration in 30 years: A survey*. arXiv:2412.13735, 2024.
>
> [2] C. Choy, J. Park, and V. Koltun. *Fully convolutional geometric features*. In *Proceedings of the IEEE/CVF International Conference on Computer Vision (ICCV)*, pp. 8958–8966, 2019.
>
> [3] H. Yu, F. Li, M. Saleh, B. Busam, and S. Ilic. *Cofinet: Reliable coarse-to-fine correspondences for robust point cloud registration*. In *Advances in Neural Information Processing Systems (NeurIPS)*, pp. 23872–23884, 2021.
>
> [4] Z. Qin, H. Yu, C. Wang, Y. Guo, Y. Peng, and K. Xu. *Geometric transformer for fast and robust point cloud registration*. In *Proceedings of the IEEE/CVF Conference on Computer Vision and Pattern Recognition (CVPR)*, pp. 11143–11152, 2022.
>
> [5] H. Chen, P. Yan, S. Xiang, and Y. Tan. *Dynamic cues-assisted transformer for robust point cloud registration*. In *Proceedings of the IEEE/CVF Conference on Computer Vision and Pattern Recognition (CVPR)*, pp. 21698–21707, 2024.
>
> [6] R. Yao, S. Du, W. Cui, C. Tang, and C. Yang. *Pare-net: Position-aware rotation-equivariant networks for robust point cloud registration*. In *European Conference on Computer Vision (ECCV)*, pp. 287-303, 2024.
>
> [7] R. Huang, Y. Tang, J. Chen, and L. Li. *A consistency-aware spot-guided transformer for versatile and hierarchical point cloud registration*. *Advances in Neural Information Processing Systems*, 37:70230–70258, 2024.
>
> [8] S. Yan, P. Shi, and J. Li. *Ml-semreg: Boosting point cloud registration with multi-level semantic consistency*. In *European Conference on Computer Vision (ECCV)*, pp. 19–37, 2025.

---

> ### Author Response · Authors · 2025-11-18
> **Response to  ''2.2. Comparisons with classical spectral descriptors (HKS/WKS).''**
>
> We sincerely appreciate the reviewer's valuable suggestions, particularly regarding the comparison with classical spectral descriptors HKS and WKS. This is indeed an important consideration that helps to more comprehensively evaluate the effectiveness of our method. As classical spectral descriptors, HKS and WKS do possess rigid motion invariance on continuous manifolds and ideal meshes. However, the weighted summation process of these kernel functions is essentially a low-pass filtering operation that smooths out the most extreme, and potentially most discriminative, details in the highest frequency components. Although WKS, by using logarithmic Gaussian kernels for selection in energy bands, better preserves high-frequency separability than HKS, it still falls short of being as "sharp" as the "raw" high-frequency eigenvectors. **This contradicts our original intention of obtaining high-frequency components.**
>
> Although we can construct high pass filters based on variants of HKS and WKS, taking the  specific high pass filter ${g_{high}}(L){\text{ }} = {\text{ }}I{\text{ }} - \beta{\text{ }}exp( - tL)$ as an example with $\beta = 0.1, t=0.5$, the Laplacian matrix $L$ is:
> $$
> L = \begin{bmatrix}
> 1.00, & -0.30, & 0.00, & 0.00, & -0.70, & 0.00; \\
> -0.30, & 1.00, & 0.00, & -0.10, & -0.60, & 0.00; \\
> 0.00, & 0.00, & 1.00, & -0.50, & 0.00, & -0.50; \\
> 0.00, & -0.10, & 0.00, & 1.00, & 0.00, & -0.90; \\
> -0.70, & -0.30, & 0.00, & 0.00, & 1.00, & 0.00; \\
> 0.00, & -0.10, & -0.90, & 0.00, & 0.00, & 1.00
> \end{bmatrix}
> $$
> Then, the high-pass filter $g_{\text{high}}(L) $  is:
> $$
> g_{\text{high}}(L) = \begin{bmatrix}
> 0.94, & -0.12, & -0.10, & -0.10, & -0.14, & -0.10; \\
> -0.12, & 0.94, & -0.10, & -0.11, & -0.13, & -0.10; \\
> -0.10, & -0.10, & 0.94, & -0.13, & -0.10, & -0.13; \\
> -0.10, & -0.11, & -0.10, & 0.94, & -0.10, & -0.16; \\
> -0.14, & -0.12, & -0.10, & -0.10, & 0.94, & -0.10; \\
> -0.10, & -0.10, & -0.11, & -0.16, & -0.10, & 0.94
> \end{bmatrix}
> $$
> Through comparative analysis, we observe that:
>
> 1. **Nonlinear Response Curve**: The high-frequency amplification effect is uneven, and the overall scaling effect is relatively smooth.
> 2. **Loss of Matrix Sparsity**: The transformation from sparse to non-sparse matrix prevents sparse processing and increases computational complexity.
>
> Our method offers the following advantages over HKS and WKS:
>
>    1.**High Computational Efficiency**: Since $L$ maintains sparsity, each point in the point cloud only needs weighted combination with neighbors, enabling local neighborhood-based computation.
>
>    2.**Significant High-Frequency Enhancement**: More pronounced high-frequency amplification effect.

---

> ### Author Response · Authors · 2025-11-18
> **Response to  ''2.3. Discriminative power of the descriptor.''**
>
> We sincerely thank the reviewer for this valuable and highly constructive comment. The issue of verifying the descriptor's discriminative power that you raised is crucial.
>
> In response to your concern, we have supplemented the manuscript with a **visualization analysis of the invariance of high-frequency components under rigid transformations**. Specifically, we used the Bunny dataset from the Stanford 3D Scanning Repository as an example, downsampled it to 2,000 nodes, and constructed the high-frequency component descriptor ${\mathbf{h}\_m} = \| [\mathcal{L} \mathbf{y}]\_{m,:} \||_2$ for each node $m$ , where $\mathcal{L}$ is the Laplacian matrix and $\mathbf{y}$ represents the node coordinates. After normalizing the descriptors, we visualized the distribution of high-frequency components for both the original and transformed point clouds in **Fig. 2** in the manuscript. From Fig. 2, it can be clearly observed that although the point cloud undergoes rigid transformations, the distribution pattern of its high-frequency components remains consistent. This visually validates the theoretical results in **Theorem 1** regarding the rigid transformation invariance of our descriptor.

---

> ### Author Response · Authors · 2025-11-19
> **Response to  ''2.1. Novelty''.**
>
> Thank you very much for your valuable feedback. We have carefully reviewed your feedback on innovation and provided a more in-depth analysis and elaboration of our core contributions.  We fully understand the concerns of the reviewers and will rewrite the contribution section in the new manuscript to make our key innovations clearer and more specific. We believe that the main innovation of this study lies not in proposing a completely new registration paradigm, but in fundamentally innovating the classic CPD framework from a new perspective of graph signal processing, systematically addressing its two long-standing core flaws:
>
> 1.Isotropic covariance problem: Traditional GMM uses ${\sigma ^2}I$ as the covariance, which assumes that the point cloud varies uniformly in all directions, completely ignoring local surface structure and geometric anisotropy.
>
> 2.Inefficient problem of uniform prior search: Without prior knowledge, CPD assumes that each point may correspond to any point, resulting in a computational complexity of $\mathcal{O}(mn)$, which is its biggest bottleneck. To our knowledge, there is currently no method based on CPD that considers the issue of prior probability.
>
> Our **key innovations** can be summarized in the following three aspects:
>
> 1. **Representation Dimensionality Expansion from "Points" to "Graph Signals" with Local Graph Modeling**. Although our approach builds upon the CPD framework, we move beyond merely using 3D coordinates and normal vectors. Instead, we construct a high-dimensional graph signal for each node and establish a corresponding local graph based on this representation. While previous works primarily operate at either the point level or global graph level, our method creates a high-dimensional descriptor graph for each point, presenting a novel representation learning perspective. This is no longer merely a "descriptor" but rather a structured local feature representation.
> 2. **Employing the pseudo-inverse of  Graph Laplacian as Structured Covariance to Incorporate Geometric Meaning**. We replace the isotropic covariance $σ²I$ with the pseudo-inverse of the local Laplacian matrix and utilize the Mahalanobis distance $||·||_{L_m}$ in the objective function. This approach not only requires spatial proximity between corresponding points but also demands consistency in the variation trends of their multi-attribute joint representations within the local graph structure. This is a capability that traditional CPD methods cannot possess.
> 3.  **Constructing Efficient and Theoretically Grounded Priors Through High-Frequency Invariance**. We introduce an effective prior probability construction method based on GSP invariance for the first time in the CPD framework.  This fundamentally differs from simply using descriptors as a preprocessing step, as it forms an intrinsic component of the probabilistic model. Its theoretical validity is guaranteed by Theorem 1, while reducing computational complexity by an order of magnitude.
>
> We fully understand your point that GraphReg has also explored graph filtering. We want to distinguish the differences between our work and GraphReg here. GraphReg is a combination of **hard matching and physical models**, while GraphCPD is a combination of **soft matching and probabilistic models**. The physical model relies on heuristic force models and simulated annealing to find the global optimum, which is a stochastic optimization method. Its convergence to the global optimum has no theoretical guarantee and heavily relies on parameters such as initial temperature and cooling plan. The hard matching method may have significant errors, which can greatly affect subsequent iterations after one-on-one matching errors in the early stage. The performance of the CPD based method (ours) is better. In addition, physical methods are more sensitive to outliers.  GraphReg uses geometric invariants for hard matching (Equation 13). Once matching errors occur due to noise, low overlap, or symmetry, incorrect attractive or repulsive forces will be generated, which will directly pull the system in the wrong direction and may lead to catastrophic failure. GraphCPD uses soft allocation, where the contribution of each point to the overall objective function is weighted. The correspondence between individual errors does not have a decisive impact on the overall transformation estimation, making it inherently more robust to outliers and incorrect matches. Although we all use knowledge of graph theory, **GraphReg focuses on constructing high pass filters to detect outliers** and does not involve them in the registration process. Our method focuses on constructing high-frequency components to directly participate in the registration process and solves the problem of slow matching speed in the CPD method.

---

### Official Review · Reviewer_nxEK · 2025-11-01

**Soundness:** 2
**Presentation:** 2
**Contribution:** 2
**Rating:** 4
**Confidence:** 3

**Summary:**

The paper proposes a new 3D point cloud registration method, leveraging some concepts from graph signal processing (GSP). In particular, for each 3D point, a local graph Laplacian is constructed, where the edge weights are computed as an exponential function of the differences in 3D locations and estimated surface normals (equation 3). The sample of a graph signal is defined by stacking the point coordinate with the surface normals, and high-frequency components (equation 2). High frequencies are computed using simple polynomials of the normalized graph Laplacian matrix $\mathcal{L}$ interpreted as the graph shift operator (GSO). The rigid transformation is estimated via an EM algorithm.

**Strengths:**

Leveraging GSP concepts in the 3D point cloud registration problem appears to be new.

**Weaknesses:**

1. The writeup is not that clearly described with terminologies that are not easy to understand. For example, what is meant by "stronger geometric significance" (pg.4)? Graph signals $\mathbf{s}_n$ and $\mathbf{s}_m$ discussed in page 3 are not defined till page 4. What is meant by the subscript $\mathbf{L}_m$ in equation (1)? If the authors mean $\ell_2$-norm, like $(\mathbf{y} - \mathbf{x}^\top \mathbf{L}_m (\mathbf{y} - \mathbf{x})$, then the fidelity term is pretty standard in signal restoration and not novel. The employed GSP terminology is not consistent with the GSP literature. A GRAPH SIGNAL $\mathbf{x} \in \mathbb{R}^N$ is typically defined as a $N$-dimensional signal, one scalar-valued sampled $x_i$ for each node $i$. Hence, $\mathbf{s}_n$ is NOT a graph signal, but a vector-valued sample at node $n$. It is not clear why $\epsilon$ is needed for "numerical stability" to define $\tilde{\mathbf{L}}_m$ (pg. 3). It is not clear why the particular "high-pass" filters in equation (4) are used, given none of them are ideal or approximately ideal high-pass filters. They seem to be defined in a ad-hoc manner.

2. Whether high-frequency components highlight local variations depend entirely on how the edge weight are defined. If the edge weights are defined, like equation (3), where expected signal variations across nodes are ALREADY encoded as small positive edge weights, then high-frequency components DO NOT highlight local variations. High frequencies actually are components that are CONTRARY to encoded pairwise similarities in the edge weights, for example, big variations across large edge weights. Big local pairwise differences across a very small weight edge (or a negative edge) does not constitute high frequencies. There is a misunderstanding here.

3. Graph Laplacian matrix is conventionally interpreted as the INVERSE of the covariance matrix also known as the precision matrix. For example, given an empirical covariance matrix $\mathbf{C}$, the sparse inverse covariance matrix $\mathbf{P}$ of a Gaussian Markov Random Field (GMRF) is often computed using graphical lasso (GLASSO) or its variants. In this proposal, however, the locally constructed Laplacian is replacing the covariance matrix in a Gaussian Mixture Model (GMM). This is a mismatch.

4. The definition of a sample of a graph signal (rather than the graph signal as written) as vector quantity in equation (2) is highly unusual, because the attributes defined in this vector-valued sample $\mathbf{s}_m$ are fundamentally different quantities in different scales. (Maybe this is why the surface normal is "scaled" in a strange manner? Not clear from the text.) So it is highly unlikely that the SAME Laplacian matrix can describe pairwise similarities for all the attributes in the vector-valued samples.

5. The experimental results in Table 1 etc are not dramatically better than previous works.

6. There are no Appendices attached to the manuscript, despite numerous mentions in the paper.

**Questions:**

1. Why is the particular edge weight definition in equation (3) employed? In Dinesh et al. 2022, the defined edge weights were defined as such for 3D point cloud geometry restoration (denoising, etc). An edge weight tends to zero when two points are far in Euclidean distance (and thus bears no similarity) OR when two points have orthogonal surface normals (e.g., different sides of a table, and thus one point cannot help the other in denoising). It's not clear why the definition is reused here for registration.

2. Why are the high-pass filters so defined in equation (4), given none of them are ideal high-pass filters? One can approximate an ideal high-pass filter, with a target cutoff frequency, using Chebyshev or Lanczos approximation with polynomials of a graph shift operator (GSO). Why is this not done?

3. Why is the invariance of high-pass components even important in this scenario?

4. Where are the Appendices?

---

> ### Author Response · Authors · 2025-11-19
> **Response to "1.1. Clarify the description in the manuscript".**
>
> We sincerely thank the reviewer for carefully reading our paper and providing highly constructive comments. Regarding the many detailed issues you pointed out, we will carefully revise and improve them in the revised manuscript.
>
> 1. **Stronger geometric significance.**  We apologize for the earlier version where the appendix was only included in the supplementary materials. Detailed derivations are provided in **Appendix A.2** （In the original version, we placed the appendix in the supplementary materials. In the next version, we will place the appendix after the main text. The enhanced geometric implications specifically refer to:
>
>    ① ensuring the minimized distance between the transformed source signal $\mathcal{T}({{\mathbf{s}}\_n})$ and the target signal ${{\mathbf{s}}\_m}$.
>
>    ② enforcing graph smoothness of the transformed source signal  on $\mathcal{T}({{\mathbf{s}}\_n})$ the corresponding graph $\mathbf{G}_m$.
>
>    ③ imposing consistency in the local variation trends between ${{\mathbf{s}}\_m}$ and  $\mathcal{T}({{\mathbf{s}}\_n})$ .
>
> 2. **The meaning of subscripts in equation (1).** The $\mathbf{L}_m$ represents the Laplacian matrix of the local graph $\mathbf{G}_m$ corresponding to the $m$-th node in point cloud $\mathcal{Y}$. Although fidelity terms based on graph smoothing are standard practice in the field of signal restoration, traditional methods often model the entire point cloud as a single graph structure and build a global norm based on this.  However, such methods often overlook the local differences between nodes when performing point cloud matching. To this end, we propose a dimensionality enhancement strategy. Specifically, we did not just focus on coordinates and normals, but instead constructed a high-dimensional graph signal for each node and based on it, created a local graph to capture the differences between points more finely.
>
> 3. **Comparison between GSP terminology used and GSP literature.** Traditional methods often consider multiple observed entities (such as sensors) as graph nodes, and their observed values are graph signals. In contrast, we consider the multidimensional signal  on a single point cloud node $m$ as a graph signal ${s\_m} \in {\mathbb{R}\^ 9}$, with each one-dimensional attribute is mapped to a virtual node. This virtual node is different from the nodes in the point cloud, but the graph signal concept used is consistent. This dimensionality enhancement processing essentially constructs a new local graph that describes the internal structure of the signal, in order to reflect the differences between different points in the same point cloud and assist in registration. Thank you very much to the reviewer  for your suggestion, and we will make this clear in the manuscript.
> 4. **The parameter numerical stability $ \varepsilon {I} $ mentioned when defining ${L\_{m}}$.**  The reasons for introducing the manuscript are as follows:
>
>    ① The component density function can represented as
>
>    $$p({{\mathbf{s}}\_n}\mid m) = {c\_m}\exp \left( { - \tfrac{1}{{2{\sigma ^2}}}\left\| {{{\mathbf{s}}\_m} - \mathcal{T}({{\mathbf{s}}\_n})} \right\|_{{L\_m}}^2} \right) \propto \mathcal{N}\left( {\mathcal{T}({{\mathbf{s}}\_n}),L\_m^+ } \right)$$
>
>    The covariance matrix $L_m^+$  is the pseudo inverse of the Laplacian matrix  $L_m$. Since the Laplacian matrix has zero eigenvalues, its pseudo inverse is infinite in the corresponding direction, which can cause the covariance matrix to be singular and result in Gaussian distribution degradation.
>
>    ② In the M step of the EM algorithm, if optimized $L_m$ directly, matrices with eigenvalues very close to zero may be generated in some iterations. This may lead to ill conditioning of the Hessian matrix, affecting convergence.
>
>    ③ Small regularization $\varepsilon I$ can prevent algorithm overfitting.
>
>    ④ The introduction of  term ${\left\| {{{\mathbf{s}}_m} - \mathcal{T}({{\mathbf{s}}_n})} \right\|^2}$ can ensure the minimum distance between the transformed source signal ${\mathcal{T}({{\mathbf{s}}_n})}$ and the target signal ${{{\mathbf{s}}_m}}$.

---

> > ### Author Response · Authors · 2025-11-19
> > **Response to "1.2. High-frequency components highlight local variations."**
> >
> > Thank you for your insightful and thorough comments. We fully agree with your observation that unweighted graphs tend to yield higher variance in the spectral distribution compared to weighted graphs. As you rightly pointed out, when two points differ significantly while the edge weight is a small positive value, the high-frequency components do not excessively amplify such differences.  Constructing a weighted graph essentially imposes adaptive smoothing constraints on the graph signal, equivalent to performing "intelligent low-pass filtering" in the graph domain.  The reason why we adopt this design is precisely to effectively reduce the variance of the spectral distribution and give it a clear geometric meaning. Point clouds often contain noise and outliers, which typically manifest as very high-frequency components. If left uncontrolled, these components can dominate the spectral distribution and obscure the mid-frequency geometric features that are more meaningful for registration. By setting an appropriate kernel bandwidth $\sigma$, we can avoid over-smoothing (which would lose geometric discriminatively) while preventing extreme high frequencies from overwhelming the mid-frequency information.
> >
> > We truly appreciate your deep understanding of graph signal processing (GSP). The use of weighted graphs for extracting high-frequency information is a well-established practice in graph theory and GSP. For instance, the weighted graph Laplacian serves as the default theoretical framework in the foundational GSP paper[1]. Moreover, the FastMAC [2] adopted the same weighted graph construction method as our Eq. (3) and successfully applied it to high-frequency filtering tasks, further validating the effectiveness of this approach.
> >
> > To further investigate this matter, we specifically designed a comparative experiment between weighted and unweighted graphs based on Experiment 1 of the manuscript. The corresponding experimental settings are following Experiment 1. To ensure a fair comparison, all tests were performed with a fixed iteration number of 30 and parameter $k_{match}=100$. The results are shown below.
> >
> > *Table 1: Performance comparison between weighted and unweighted graphs (AngErr, °)*
> >
> > | Dataset   | $W_{weighted}$ | $W_{unweighted}$ |
> > | --------- | -------------- | ---------------- |
> > | Dragon    | 0.120          | 0.832            |
> > | Armadillo | 0.098          | 0.308            |
> >
> >
> >
> > [1] D. I. Shuman, S. K. Narang, P. Frossard, A. Ortega, and P. Vandergheynst, “The emerging field of signal processing on graphs: Extending high-dimensional data analysis to networks and other irregular domains,” IEEE Signal Processing Magazine, vol. 30, no. 3, pp. 83–98, 2013.
> >
> > [2] Yifei Zhang, Hao Zhao, Hongyang Li, and Siheng Chen. Fastmac: Stochastic spectral sampling of correspondence graph. In Proceedings of the IEEE Conf. on Computer Vision and Pattern Recognition (CVPR), pp. 17857–17867, 2024.

---

> ### Author Response · Authors · 2025-11-19
> **Response to  "1.3. Graph Laplacian matrix is conventionally interpreted as the INVERSE of the covariance matrix also known as the precision matrix. "**
>
> Thank you very much for pointing out this inaccuracy in our phrasing. Your insight into the connection with GMRFs is highly appreciated and absolutely correct. You are right that in the theories of GSP and GMRF Fields, the graph Laplacian matrix ( $L$ ) is more precisely interpreted as the precision matrix (i.e., the inverse covariance matrix, ${\Sigma ^{ - 1}}$), not the covariance matrix itself.
>
> We have rigorously corrected this throughout the manuscript (specifically in the abstract) by changing "the graph Laplacian matrix replaces the covariance matrix" to "**the pseudo-inverse of the graph Laplacian matrix replaces the covariance matrix**." We sincerely thank you for helping us improve the precision of our presentation.
>
> Building upon this correction, we would like to clarify that our revised formulation is, in fact, highly consistent with the GMRF framework. In our locally constructed graph, the non-zero entries ($L\_{ij}≠ 0$) in the Laplacian matrix $L$ correspond precisely to pairs of nodes $(i, j)$ that are *conditionally dependent* given all other nodes in the GMRF interpretation. This directly reflects the core "conditional independence" assumption of GMRFs. Therefore, our use of the pseudo-inverse $L^+$ to define the local covariance structure is mathematically equivalent to defining a GMRF via a sparse precision matrix.
>
> For your reference, while the textual description was inadvertently imprecise, all corresponding formulas  in our derivations have been correct from the outset. Thank you again for your rigorous review, which has allowed us to provide a clearer explanation of this point.

---

> ### Author Response · Authors · 2025-11-19
> **Response to  "1.4. The scale problem of graph signal."**
>
> Thank you for this insightful comment. Your observation regarding the vector-valued samples containing inherently heterogeneous attributes at different scales, and the doubt about whether a single Laplacian can describe pairwise similarities for all of them, is highly pertinent. We acknowledge that the description of this core design in our initial manuscript was not sufficiently clear, which may have led to misunderstanding. We have now rephrased the relevant sections for better clarity. Our response aims to clarify this issue from two perspectives:
>
> **1. Rationale for Scale Normalization:**
>
> You are correct that coordinates, normals, and high-frequency components are heterogeneous by nature. This is precisely the reason for the scale normalization step we employ prior to Equation (2). Specifically, as mentioned in Section 3.2 , we normalize the normal vectors as follows:
> $$ {{{\mathbf{\bar n}}} \_m} = \frac{{\max \left( {\mathbf{y}} \right) - \min \left( {\mathbf{y}} \right)}}{{\max \left( {\mathbf{n}} \right) - \min \left( {\mathbf{n}} \right)}}{{\mathbf{n}} \_m} $$
> This step ensures that all 9 attribute dimensions are projected into a comparable numerical range. This preprocessing is a standard practice in GSP (e.g., when signals come from different sensors) and multi-modal data fusion. Its purpose is to eliminate initial scale disparities, allowing the subsequent Euclidean-distance-based kernel to weigh each dimension's contribution to similarity fairly. The purpose is to avoid certain attribute (coordinates or normals) dominating similarity calculations due to scale differences, so that subsequent Euclidean distance based kernels can fairly balance the contribution of each dimension to similarity.
>
> **2.Rationale of the Local-Graph Laplacian:**
>
> We wish to emphasize that for each node $\mathbf{y}\_m$ in the point cloud $\mathcal{Y}$, we construct a unique, node-specific local graph $\mathcal{G}\_m$ and its corresponding Laplacian matrix $\mathbf{L}\_m$. Specifically, the local graph consist of 9 attributes (3 coordinates, 3 normal vectors, 3 high-frequency components), and edge weights measure the statistical correlation between different attribute dimensions. In **Section A.2 of the Appendix**, we explain in detail why the graph Laplace matrix has geometric explanatory significance:
>
> The $\left\| {{{\mathbf{s}}\_m} -\mathcal{T}({{\mathbf{s}}\_n})} \right\|\_{{{\tilde {\mathbf{L}}}\_m}}^2$ can be decomposed into
> $$
> \left\| \mathbf{s}\_m - \mathcal{T}(\mathbf{s}\_n) \right\|\_{\widetilde{\mathbf{L}}\_m}^2
> 	= \varepsilon \left\| \mathbf{s}\_m - \mathcal{T}(\mathbf{s}\_n) \right\|^2
> 	+ \left\| \mathbf{s}\_m - \mathcal{T}(\mathbf{s}\_n) \right\|\_{\mathbf{L}\_m}^2,
> $$
> where the first term on the right side of the equation  is to ensure that the distance between the transformed source signal $ \mathcal{T}\left({{\mathbf{s}\_n}} \right)$ and the target signal $\mathbf{s}\_m$ is minimum. Expanding the second term yields:
>
> $$
> \begin{aligned}
>  		&\left\| {{{\mathbf{s}}\_m} - \mathcal{T}({{\mathbf{s}}\_n})} \right\|\_{{{\mathbf{L}}\_m}}^2 \hfill \\
>  		=& {{\mathbf{s}}\_m}^T{{\mathbf{L}}\_m}{{\mathbf{s}}\_m} + \mathcal{T}{({{\mathbf{s}}\_n})^T}{{\mathbf{L}}\_m}\mathcal{T}({{\mathbf{s}}\_n})
>  		 - 2\sum \limits \_{i,j = 1,..,9} {{{\mathbf{W}\_m^{ij}}}\left( {{{\left[ {\mathcal{T}({{\mathbf{s}}\_n})} \right]}\_i} - {{\left[ {\mathcal{T}({{\mathbf{s}}\_n})} \right]}\_j}} \right)} \left( {{{\left[ {{{\mathbf{s}}\_m}} \right]}\_i} - {{\left[ {{{\mathbf{s}}\_m}} \right]}\_j}} \right)
>  \end{aligned}.
> $$
> The term $\mathcal{T}(\mathbf{s}\_n) ^\top \mathbf{L}\_m \mathcal{T}(\mathbf{s}\_n)$ reflects the  graph smoothness  of the transformed source signal $\mathcal{T}(\mathbf{s}\_n)$ on graph $\mathbf{G}\_m$, while the cross term quantifies the  consistency of local variation trends  between $\mathbf{s}\_m$ and $\mathcal{T}(\mathbf{s}\_n)$ over the same graph. Specifically, a larger value of the cross term indicates stronger alignment in how the two signals vary across neighboring nodes. In summary, replacing the isotropic covariance with the Laplacian matrix $\mathbf{L}\_m$ not only preserves the classical objective of minimizing the signal discrepancy between the transformed source and the target, but also incorporates graph characteristic that enhance the registration process. It imposes graph smoothness constraints on the transformed signal and  consistency constraints on the variation trends of signals over the graph structure. These properties promote geometrically meaningful and transformation-consistent correspondences, leading to improved robustness and interpretability in point cloud registration.

---

> ### Author Response · Authors · 2025-11-19
> **Response to  "1.6. The appendix."**
>
> Thank you for this meticulous reminder. We sincerely apologize for this oversight. In our previous submission, this content was placed in a separate "Supplementary Material" file, which may understandably have caused confusion during your review. As per your comment, we have now integrated all relevant content from the supplementary material into the main manuscript as a formal appendix section and will update  the PDF file for this revision. We greatly appreciate your help in improving the presentation of our work.

---

> ### Author Response · Authors · 2025-11-20
> **Response to  "1.5. Table 1 Experimental results and performance analysis."**
>
> We sincerely thank the reviewer for presenting this important viewpoint on performance analysis. We fully understand and appreciate the concerns of the reviewers. In the previous version, while our method demonstrated strong performance in terms of accuracy, it faced limitations in computational speed. **In the updated code (https://anonymous.4open.science/r/GraphCPD-801E) , we have optimized the calculations, and we believe these enhancements significantly improve the competitiveness of our approach**. Specifically, the following improvements have been made:
>
> **1.GPU Parallel Computing.** To enhance the efficiency of our method, we have reformulated the computations in the E-step of the EM algorithm as parallelizable matrix operations that can be efficiently executed on GPU hardware.
>
> **2. Neighborhood-based Probability Simplification.** We simplified the probability calculation for $P$ in the E-step by leveraging the $k\_{match}$ nearest neighbors of the highest frequency components. Specifically, for the $m$-th node, we only compute probabilities $p\_{mn}$ for nodes where $\mathbf{y}\_m \in \mathcal{C}\_n$, where $\mathcal{C}\_n$ represents the set of $k\_{match}$ closest candidates to the $m$-th node. Furthermore, we optimized the computation of the term $\left( { - \tfrac{1}{{2{\sigma ^2}}}\left\| \|{{{\mathbf{s}}\_m} -\mathcal{T}({{\mathbf{s}}\_n})} \right\|\ |\ _{{{\tilde {\mathbf{L}}}}\_m}^2} \right)$ in $P\_{mn}$.
>
> **3.CUDA Kernel Optimization.**  To further accelerate performance, we implemented custom CUDA kernels. Since GPUs are not inherently efficient at matrix rearrangement based on indices, our custom CUDA scripts specifically address this bottleneck.
>
>
>
> Moreover, in  earlier manuscript, the slightly lower accuracy compared to LSG-CPD was due to our introduction of a prior probability  $\mathbf{P}\_{mn}$ to address the limitation of traditional probabilistic based methods. The LSG-CPD is built upon the classical probabilistic based  framework, which requires computing the matching probabilities between all point pairs in the source and target point clouds, resulting in a computational complexity of O(MN).  By contrast, our method restricted each node m to match only  $k\_{match}$ nodes, resulting in a computational complexity of O(kN), where k≪M. In the earlier manuscript, this approach allowed us to achieve a 4.7× speedup at only a minor cost of 0.017° in accuracy, which holds practical significance for real-world applications.
>
> **In the revised manuscript**, considering the slow speed of traditional probability based methods, this time is unacceptable in practical applications. We re evaluated the LSG-CPD and ECMPR methods by downsampling the point set to 8000 nodes to fairly compare accuracy and runtime. The updated results are presented in Table 1 below:
>
> Table 1: Registration accuracy and runtime for Dragon and Armadillo datasets.
>
> | Method    | Dragon AngErr (°) | Dragon Time (s) | Armadillo AngErr (°) | Armadillo Time (s) |
> | --------- | ----------------- | --------------- | -------------------- | ------------------ |
> | GraphReg  | 0.403             | 4.30            | 0.284                | 5.63               |
> | CPD       | 0.624             | 96.57           | 0.597                | 97.20              |
> | LSG-CPD   | 0.155             | 0.862           | 0.138                | 0.758              |
> | FilterReg | 0.297             | 20.05           | 0.171                | 13.51              |
> | TrICP     | 0.800             | 0.50            | 0.119                | 0.55               |
> | ECMPR     | 0.203             | 133.40          | 0.173                | 137.86             |
> | GraphCPD  | **0.091**         | 2.950           | **0.095**            | 2.846              |

---

### Meta-Review · Area_Chair_zeZJ · 2026-01-06

**Summary:**

The reviewers express several concerns about clarity, correctness, novelty, and evaluation. In places the write-up is difficult to follow, with undefined or misused terminology (e.g., “stronger geometric significance,” graph signals, subscripts in equations), inconsistent and sometimes incorrect use of graph signal processing concepts, and unclear motivations for key design choices such as the need for numerical stabilization, the construction of vector-valued graph signal samples, and the ad-hoc definition of high-pass filters. The assumption that a single Laplacian can model similarities across heterogeneous, differently scaled attributes is highly questionable. From a contribution standpoint, the reviewers indicate that the method shows limited novelty, largely repackaging hand-crafted graph-Laplacian high-frequency descriptors within a standard CPD framework, with performance gains that are marginal and not clearly superior to prior work. The evaluation is weak and incomplete, missing comparisons to canonical spectral descriptors (HKS/WKS) and modern deep learning–based methods, lacking evidence of descriptor discriminativeness, omitting important baselines in several experiments, and providing insufficient justification for key parameter choices. Efficiency and scalability are also indicated as important concerns, with relatively high runtimes on small datasets and a narrow experimental scope, further limiting the practical and scientific impact of the work.

**Reviewer Concerns:**

The authors have tried to provide exhaustive comments to the reviewers and I think in part they have done a good job. For sure they have clarified the confusion about the Appendix and have provided a good explanation about the introduced concepts. The novelty is still a problem and so are the experiments. The authors have provided experiments on a new dataset which are appreciated but I do not think they can change the overall impression. The scalability issue has also been addressed at least partially but I would have preferred to have it in the manuscript from the very beginning given that this is an important aspect and the authors could have easily anticipated that the reviewers would comment directly.

**Reviewer Scores:**

I think the discussion might have improved some of the opinions of the reviewers but I doubt the overall outcome would have been different. The main problem is that all the initial scores were rather negative and the concerns were real. This has been acknowledged also by the authors in their rebuttal. I believe reviewer nxEK would have been happy with the provided clarifications but still the concerns about the minor performance improvement would have remained. The reviewer SbMu was mainly concerned about the limited novelty and the comparison with several other approaches. While the latter has been well-addressed by the authors the novelty issue has been only partially considered. The reviewer VvGA was mostly concerned about the experiments and the authors have addressed this in a satisfactory manner but the feeling is that the amount of chances required in the revision is quite substantial. Overall, I think the authors have done a good job in their rebuttal but the starting point was too low to change the final decision.

---

### Decision · Program_Chairs · 2026-01-26

Reject